# Accurate integration of single-cell DNA and RNA for analyzing intratumor heterogeneity using MaCroDNA

Mohammadamin Edrisi [1] ✉, Xiru Huang[1], Huw A. Ogilvie [1] ✉ & Luay Nakhleh [1] ✉

Cancers develop and progress as mutations accumulate, and with the advent of single-cell DNA and RNA sequencing, researchers can observe these mutations and their transcriptomic effects and predict proteomic changes with remarkable temporal and spatial precision. However, to connect genomic mutations with their transcriptomic and proteomic consequences, cells with either only DNA data or only RNA data must be mapped to a common domain. For this purpose, we present MaCroDNA, a method that uses maximum weighted bipartite matching of per-gene read counts from single-cell DNA and RNA-seq data. Using ground truth information from colorectal cancer data, we demonstrate the advantage of MaCroDNA over existing methods in accuracy and speed. Exemplifying the utility of single-cell data integration in cancer research, we suggest, based on results derived using MaCroDNA, that genomic mutations of large effect size increasingly contribute to differential expression between cells as Barrett's esophagus progresses to esophageal cancer, reaffirming the findings of the previous studies.

Over the last decade, single-cell sequencing (SCS) has grown dramatically. SCS can now provide data with much higher throughput and quality with a diverse range of protocols that can assay single cells' genomic, transcriptomic, epigenomic, and proteomic contents[1–6]. These technologies have changed our understanding of different fields of biology, including developmental biology, immunology, microbiology, and cancer biology.

SCS technologies can shed light on our understanding of the genotype-phenotype relationship. In particular, single-cell multi-omic assays sequencing RNA and DNA from the same single cells are of great potential to unravel the underlying mechanisms by which genomic alterations, including single-nucleotide variations, copy number aberrations (CNAs), and other structural variations, might impact transcriptomic programs at the single-cell level. This would lead to further understanding of clonal development in healthy and diseased tissues, especially in cancer therapies where drug responses are affected in part by phenotypes induced by genomic mutations[7].

Despite being ideal for such studies, SCS technologies that measure both DNA and RNA from the same cells, such as G&T-seq[8], DR-seq[9], and scTrio-seq[10], are of low throughput and face scalability issues. On the other hand, high-throughput sequencing technologies such as 10X genomics single-cell RNA sequencing (scRNA-seq)[11] or Direct Library Preparation (DLP)[12] can measure either genomic or transcriptomic content of single cells imposing the computational challenge of associating cells across different modalities. Among such computational problems, the cell association problem in high-throughput SCS data where for each single cell, either CNA measurements or gene expression values are provided is relatively less studied. There is a plethora of methods developed for the integrative analysis of SCS multi-omic data, which mostly integrate the single-cell chromatin accessibility—scATAC-seq—or DNA methylation measurements with the gene expression data from scRNA-seq such as[13–20], to name a few. However, there are yet, a handful of methods that specifically address the cell association problem between scDNA-seq and scRNA-seq data, such as clonealign[21], Cardelino[22], CCNMF[23], and SCATrEx[24]

[1]Department of Computer Science, Rice University, Houston, Texas, USA. ✉e-mail: edrisi@rice.edu; huw.a.ogilvie@rice.edu; nakhleh@rice.edu

among which clonealign, CCNMF, and SCATrEx are developed for the integration of copy number and gene expression data. In the following, we review clonealign, CCNMF, Seurat[19], and SCATrEx.

clonealign[21] is a statistical method aiming to assign single cells whose gene expression values are provided by scRNA-seq to cancer clones inferred from low-coverage single-cell CNA data. Here, the clones are the clades obtained by defining a cut-off on a phylogenetic tree reconstructed from CNA data. clonealign's dosage compensation function is predicated on the assumption that an increase (decrease) in the copy number value of a gene would yield a higher (lower) gene expression value at that gene. This assumption was established based on the previously observed relationship between the genes' expressions and their corresponding copy number values in bulk and single-cell assays[8,9,25,26]. Given the clone-specific copy number profiles, clonealign predicts the assignment of the cells measured using scRNA-seq to cancer clones measured using scDNA-seq by approximating the posterior probability distribution of the data given the method parameters using variational inference[27].

Seurat's integration method[19] is a manifold alignment algorithm to solve the integration of multiple modalities. In manifold alignment methods, it is assumed that the data points—cells in this context—from the two modalities share a low-dimensional space where each cell can be represented by a low-dimensional vector—interpreted as the underlying biological state of the cell. The cells from the two domains whose vector representations are either identical or adjacent to each other in the manifold are identified as the associated cells. To establish such a shared manifold between two different modalities, Seurat adopts canonical correlation analysis (see, e.g., ref. 28). This dimensionality reduction step is followed by the identification of mutual nearest neighbors—or anchors—across the two datasets. Anchors are the pairs of single cells, each from a different dataset, that neighbor each other in the shared low-dimensional space. The anchors are filtered, scored, and finally utilized for batch correction and further downstream analyses.

Both clonealign and Seurat are map-to-reference inference methods, meaning that the data from one domain is mapped to the clones inferred from another domain. The reference may differ in such methods. For example, in clonealign, the copy number profile of each clone is the reference, while in Seurat, the scRNA-seq data on the latent manifold is the reference. Since the choice of the reference in these methods might introduce systematic bias to the method[23], a reference-free method, CCNMF[23], was put forward. CCNMF applies a non-negative matrix factorization technique (see refs. 29,30 for details about this technique) to the scRNA-seq and scDNA-seq data. Specifically, for each of the two SCS datasets, CCNMF infers a pair of matrices, including the cluster (clonal) centroids in a low-dimensional space as well as the soft clonal assignment of the single cells to their clones. The low-dimensional representations of the DNA and RNA clones are coupled by incorporating prior information about the relationship between CNAs and gene expression values, such as the information acquired by the linear regression modeling of CNAs and gene expression values measured from the paired RNA and DNA bulk sequencing data. Performing this coupling technique on the cluster centroids of the two datasets yields a co-clustering on both datasets. CCNMF's unique co-clustering technique differentiates it from the other existing methods, such as clonealign and Seurat, as CCNMF can infer clones by itself, it does not accept the predefined clones as input. A more recent method is SCATrEx[24]. Given a CNA clonal tree, SCATrEx maps scRNA-seq cells to the CNA clones to obtain an augmented tree while accounting for two types of possible gene expression variations, including the gene expression variations influenced by clonal copy number changes along the tree, and the variations not related to the underlying clonal tree structure such as epigenetic events. The authors used discrete moves to search for new tree structures and employed mean-field variational inference to score each augmented tree.

Here we report on a method for the inference of cell association in SCS multi-omic data with CNA and gene expression measurements. Inspired by clonealign's underlying assumption that the gene expression value of a gene is proportional to the corresponding CNA value at that gene, we hypothesize that the Pearson correlation coefficients between the gene expression measurements of scRNA-seq cells and the CNA profiles of scDNA-seq cells could be an effective metric for the cell association inference. We applied our method, MaCroDNA (Mapping Cross Domain Nucleic Acid) to a scTrio-seq2 dataset[31]. We exploited the partially known ground truth information in this dataset to evaluate the performance of the methods for integration of independent single-cell gene expression and copy number data directly on empirical data. By taking advantage of the available ground-truth information and using a variety of clustering techniques, we measured the accuracy of clonealign, Seurat, and MaCroDNA in assigning the scRNA-seq cells to the scDNA-seq clones as well as predicting the clonal prevalences of the scRNA-seq cells. To demonstrate the insight that may be gained from integrated DNA and RNA single-cell data, we applied MaCroDNA to a previously published dataset from a study of Barrett's esophagus (BE), where biopsies were taken from multiple patients and individual cells from those biopsies sequenced for either DNA or RNA[32]. BE is a metaplasia presumed to be caused by gastro-esophageal reflux disease, where the squamous epithelium of the esophagus is replaced by metaplastic epithelium with intestine-like goblet cells[33]. Non-dysplastic BE (NDBE) may progress to low-grade dysplasia (LGD), high-grade dysplasia (HDG), or esophageal adeno-carcinoma (EAC), although the exact relationship of this progression in terms of cellular descent remains uncharacterized[34]. While a diagnosis of EAC is devastating with dismal prospects for survival, most BE do not progress to EAC, and predicting when EAC will develop is very difficult[35]. To make useful predictions and improve patient outcomes, we must first understand why progenitor cells evolve into EAC or dysplastic BE instead of NDBE. Our MaCroDNA-based results suggest that in comparison to NDBE and LGD, HGD and EAC are associated with genomic mutations of large effect size, which are heterogeneously present within individual biopsies. This finding aligns with previous studies suggesting that the copy number changes are good predictors of progression from Barrett's esophagus to esophageal adenocarcinoma[36–38].

## Results

### Overview of MaCroDNA

The inputs of MaCroDNA include the scRNA-seq gene expression read count tables (or their log-transformed values) and the scDNA-seq absolute copy numbers (or their log-transformed values) of the single cells sequenced from the same tissue (see Fig. 1a and also Section "MaCroDNA"). Given these inputs, for each cell in the scRNA-seq data, MaCroDNA identifies the best corresponding cell in the scDNA-seq data. Here, our main criterion for the best correspondence is the Pearson correlation coefficient between the paired cells. More specifically, MaCroDNA seeks an assignment that maximizes the sum of the Pearson correlation coefficients between the paired cells (see Section "MaCroDNA for $N_G \leq N_C$").

Supposing the single cells are sampled from the same tissue, we assume that they share the same underlying clones, and therefore, the clonal prevalences in the two datasets are nearly identical. To respect such an assumption, MaCroDNA avoids assigning any two scRNA-seq cells to the same scDNA-seq cell as much as possible. To elaborate more on how MaCroDNA accounts for this constraint, let us consider the two following cases:

- When the number of the scRNA-seq cells is smaller than the number of the scDNA-seq cells, MaCroDNA identifies a unique scDNA-seq cell for each scRNA-seq cell by viewing the correspondence problem as an instance of maximum weighted bipartite matching problem. This is a well-known problem that is

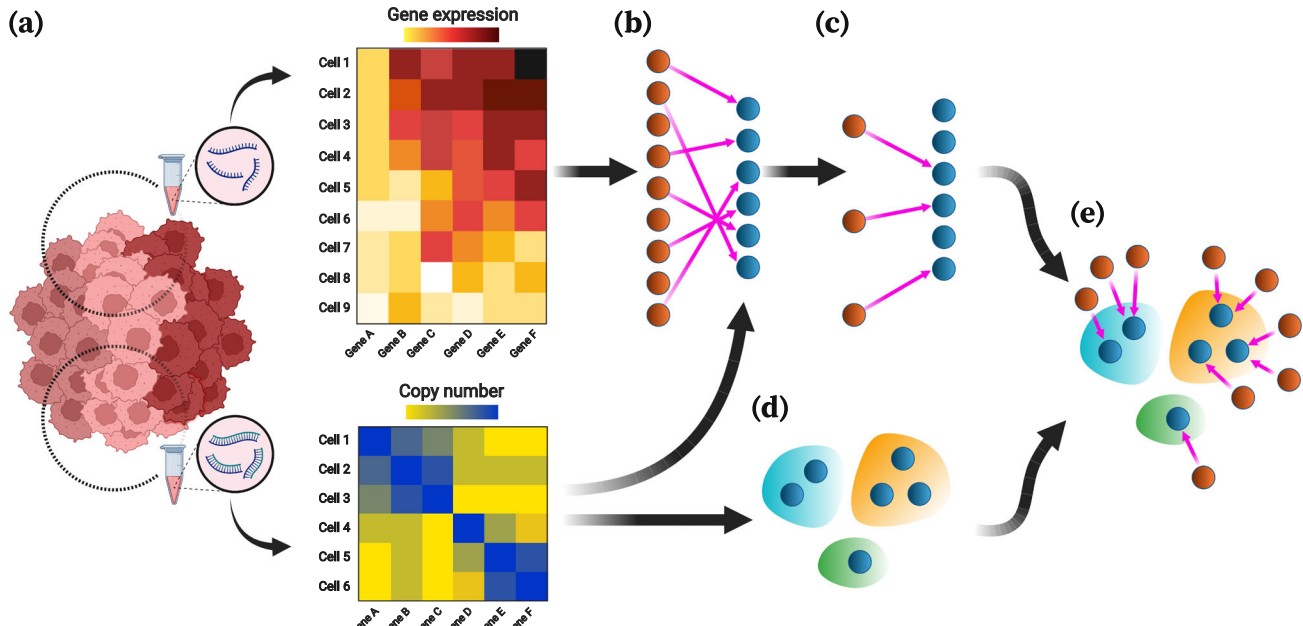

**Fig. 1 | Overview of MaCroDNA. a** The input of MaCroDNA consists of the scRNA-seq gene expression read count tables (or their log-transformed values) and the scDNA-seq absolute copy numbers (or their log-transformed values) that are supposedly obtained from the same tissue. Distinct clones are distinguished by color saturation. **b** Given the gene expression and copy number matrices (intensity of pixel colors is proportional to copy number/gene expression values), MaCroDNA identifies the assignment of the scRNA-seq cells (dark orange circles) with gene expression values to the scDNA-seq cells (dark blue circles) with copy number values. When the number of the scRNA-seq cells is higher than the number of the scDNA-seq cells (as in the above example), MaCroDNA infers the correspondence between the cells (shown by pink arrows) by solving a series of maximum weighted bipartite matching problems (in this example, two steps are required). In the first step, only six scRNA-seq cells (equal to the number of the scDNA-seq cells) are assigned to the scDNA-seq cells such that no two scRNA-seq cells are paired with the same scDNA-seq cell, and the sum of the Pearson correlation coefficients between the pairs is maximized. **c** The scRNA-seq cells whose correspondences were identified in the last step are removed. Next, the remaining three scRNA-seq cells are assigned to the best scDNA-seq cells in a one-to-one fashion according to the same correlation-based criterion. **d** The clones are inferred from the scDNA-seq data using an algorithm of choice (the clones are represented by blue, orange, and green bubble shapes). **e** Given the cell-to-cell correspondences and the clonal assignment of the scDNA-seq cells, we can assign the scRNA-seq cells to the scDNA-seq clones. The clonal assignment of a scRNA-seq cell is that of its corresponding scDNA-seq cell. This figure was created with BioRender.com.

polynomially solvable[39] (see Section "MaCroDNA for $N_G \le N_C$" for more details).

- When the number of the scRNA-seq cells is larger than the number of the scDNA-seq cells, some of the scRNA-seq cells would inevitably be matched to the same scDNA-seq cells. For this case, we devised a heuristic algorithm that breaks down the problem into multiple maximum weighted bipartite matching problems and solves them, one by one, in a sequential manner (Fig. 1b, c demonstrates an example of this case. Also see Section "MaCroDNA for $N_G > N_C$").In both cases, MaCroDNA returns as output a binary correspondence matrix whose entries indicate whether two cells (one from scRNA-seq and the other from scDNA-seq data) are matched or not.

Using the above correspondence matrix, we assign the scRNA-seq cells to the clones identified from scDNA-seq data. First, we perform a clustering technique of choice on the scDNA-seq cells to obtain the clonal assignments (Fig. 1d). Given the clonal assignment of scDNA-seq cells, the clonal assignment of a scRNA-seq cell will be that of its corresponding scDNA-seq cell (Fig. 1e).

In the following, we will describe our scheme for evaluating the performance of MaCroDNA, Seurat, and clonealign when applying them to an empirical dataset.

**Evaluating MaCroDNA on a colorectal cancer scTrio-seq dataset**
To assess the performance of MaCroDNA and the other methods, we applied them to a colorectal cancer (CRC) dataset originally studied by Bian et al.[31]. The CRC dataset consists of the scRNA-seq and scDNA-seq data of several patients. We decided to use three of these patients,

namely CRC04, CRC10, and CRC11[31]. The remaining patients were excluded from our analysis because their transcriptomic data were obtained using a different library preparation protocol or their cell amount was insufficient (see "Description of CRC dataset").

The CRC dataset is particularly helpful for our evaluation purposes as it contains some cells for which both RNA and DNA measurements are provided. These cells can readily be used in our experiments as the ground truth data. In all our experiments, we exploited this ground truth information to evaluate the performance of the methods (see below). Of note, all the methods (including MaCroDNA) were oblivious to this information at the time of inference, and we used it only for the assessment. Given the limited ability to produce realistic synthetic data, this study assesses the accuracy of methods for integrating DNA and RNA data using this empirical dataset. While simulation-based analyses are widely used and useful, they cannot match the realism of this approach and results from empirical datasets without any ground truth can only be described rather than measured for accuracy.

Table 1 provides information on the number of cells that are provided with RNA data, DNA data, and both RNA and DNA data for each patient.

In the following, we will describe the experiments we conducted on MaCroDNA, Seurat, and clonealign. Through these experiments, we sought to answer the two following questions:

1. **Cell-to-clone assignment accuracy**: Given a clonal assignment identified on the scDNA-seq cells using a clustering algorithm, what is the accuracy of a method in assigning the scRNA-seq cells to the scDNA-seq clones? We measured the accuracy only for the

**Table 1 | Cell counts from the colorectal cancer dataset**

| Patient ID | RNA[a] | DNA[a] | Both |
|------------|--------|--------|------|
| CRC04 | 93 | 93 | 57 |
| CRC10 | 85 | 123 | 69 |
| CRC11 | 192 | 249 | 174 |

[a]Inclusive of cells with both RNA and DNA.

scRNA-seq cells whose joint DNA measurement is also provided. For these scRNA-seq cells, we know their true corresponding scDNA-seq cell. Following this idea, the assignment of such scRNA-seq cell to a scDNA-seq cell is considered correct if the predicted scDNA-seq cell is the true corresponding scDNA-seq cell or if both the predicted and true scDNA-seq cells belong to the same clone according to the results of the clustering algorithm.

2. **Predictive accuracy for clonal prevalences**: According to the ground truth information and the clonal assignment of the scDNA-seq cells (again, provided by a clustering algorithm of choice), one can count the number of the scRNA-seq cells that are assigned to each of the scDNA-seq clones. While such counts can serve as the true clonal prevalences in the scRNA-seq data, we can obtain the predicted clonal prevalences by counting the number of the scRNA-seq cells that are assigned to the scDNA-seq clones by a method. In particular, we are interested in understanding the correlation between the true and predicted clonal prevalences for each clone.

It is to be noted that answering the above questions reveals two distinct characteristics of a method's performance. A method could be very inaccurate in mapping the scRNA-seq cells to the scDNA-seq clones while being faithful to the proportion of clones in the scRNA-seq data or vice versa.

We first inferred the scDNA-seq clones to measure both the cell-to-clone assignment accuracy and the predictive accuracy for clonal prevalences. Here, we chose two different clustering techniques, namely intNMF[40] and agglomerative clustering. Testing the integration methods under different clustering results leads to a better understanding of the robustness of the methods' performance to the choice of clustering algorithm.

In addition to the clustering algorithm, data transformation is another contributing factor in clonal inference. The original copy number values in the scDNA-seq datasets are in the form of non-negative integers. Along with the original copy number values, we used the log-transformed copy number values for the clonal inference as well. Log transformation is a commonly used transformation for the genomic data[41–45]. We transformed the data using $\log(x+1)$, where $x$ is the original copy number value and $\log(\cdot)$ denotes the natural logarithm.

Given the original and log-transformed data, we performed intNMF and agglomerative clustering techniques on each patient's scDNA-seq data separately. intNMF's clustering algorithm provides the optimal number of clusters along with the clonal assignments for the single cells (see "Clonal inference by intNMF"). intNMF identified two clusters in each of the patients, CRC04, CRC10, and CRC11, using the original copy number values as input. Here, in both patients, CRC10 and CRC11, the largest clone contained more than half of the patient's cells, while CRC04 had relatively balanced clusters (see Supplementary Fig. S16). Using the log-transformed data, intNMF inferred two clusters in CRC04 and three clusters in each of the patients CRC10 and CRC11 (Supplementary Fig. S18). Again, while CRC04 had two balanced clones, CRC10 and CRC11 had one or two dominant clones.

Unlike intNMF, agglomerative clustering does not infer the optimal number of clones. Instead, it takes a user-specified value for the number of clusters as input. We ran agglomerative clustering to identify four clusters in each patient independently. This resulted in 12 clusters in total across all patients. Next, we merged the clusters one by one until we reached two clusters in each patient (see "Clonal inference by agglomerative clustering" for more details on the merging procedure). By doing so, we obtained the clonal assignments at different resolutions, ranging from six to 12 clusters in total.

After inferring the clusters under different configurations of the clustering method and data transformation, we assessed the performance of MaCroDNA, clonealign, and Seurat for the cell-to-clone assignment and predicting the clonal prevalences. Although Seurat was introduced for the data integration between different scRNA-seq assays or between scRNA-seq and scATAC-seq datasets, we decided to incorporate it into this study as the exemplar of the commonly used methods in SCS multi-omic analyses. For CCNMF, we decided to assess its performance separately since it did not fit into our evaluation scheme (due to not accepting the predefined clones as its input). Even on this basis, its performance was poor, therefore we excluded its results from our main analysis and included them in Supplementary Figs. S19 and S20. To better understand the complexity of the problem and assess the methods' performance, we devised a random baseline that assigns a scRNA-seq cell to a scDNA-seq cell randomly sampled (with replacement) from the pool of scDNA-seq cells in the same patient (see "Random baseline" for the details).

Prior to running each method for integrating the scDNA-seq and scRNA-seq data, we preprocessed the CRC patients' data according to the requirements of each method as well as the preprocessing strategies that we considered (for the details on the specific preprocessing strategy applied to each method, see "Preprocessing on CRC dataset"). Since clonealign requires the clone-specific copy number profiles as input, we provided the median of all copy number profiles in a cluster as the clone-specific profile to clonealign. We ran Seurat and clonealign with their default parameters. Note that MaCroDNA does not require any user-specified parameters as the input. In all experiments related to cell-to-clone assignment and prediction of clonal prevalences (including the results in Supplementary Figs. S1–S21), we applied clustering algorithms to all DNA cells but only used the cells with both DNA and RNA data for accuracy measurement.

The accuracy results of the methods (including the baseline) for the cell-to-clone assignment are shown in Fig. 2. In the panels for agglomerative clustering (Fig. 2a, b), we have demonstrated the results of agglomerative clustering with the highest resolution, i.e., four clusters per patient. As illustrated in Fig. 2, MaCroDNA had the lowest variance and highest median value among all methods in all four different configurations. Not only did MaCroDNA achieve the highest accuracy, but it also was robust to the choice of clustering method and data transformation. In contrast, clonealign's performance was affected by these two factors. The median of clonealign's accuracy was lower than that of the baseline except for the combination of agglomerative clustering and original data, where it is comparable to that of the baseline (Fig. 2a). Compared to clonealign, Seurat showed a relatively better performance. However, its median was less than or comparable to that of the random baseline for the choices of agglomerative clustering and original data (Fig. 2a) and intNMF and log-transformed data (Fig. 2d). We would like to highlight that we examined the performance of clonealign under two preprocessing procedures and presented clonealign's best results here. The results of clonealign under both preprocessing strategies for cell-to-clone assignment are illustrated side-by-side in Supplementary Fig. S21 in Supplementary Information, Section 2.5, "Investigating clonealign's accuracy by adding pseudocount values to the inputs". Additionally, we explored the impact of various CRC data preprocessing strategies, data transformation for clustering, the clustering algorithm, and clustering resolution on the performance of these methods (Supplementary Figs. S1–S18). Under these diverse scenarios, MaCroDNA consistently demonstrated robustness and the highest level of accuracy.

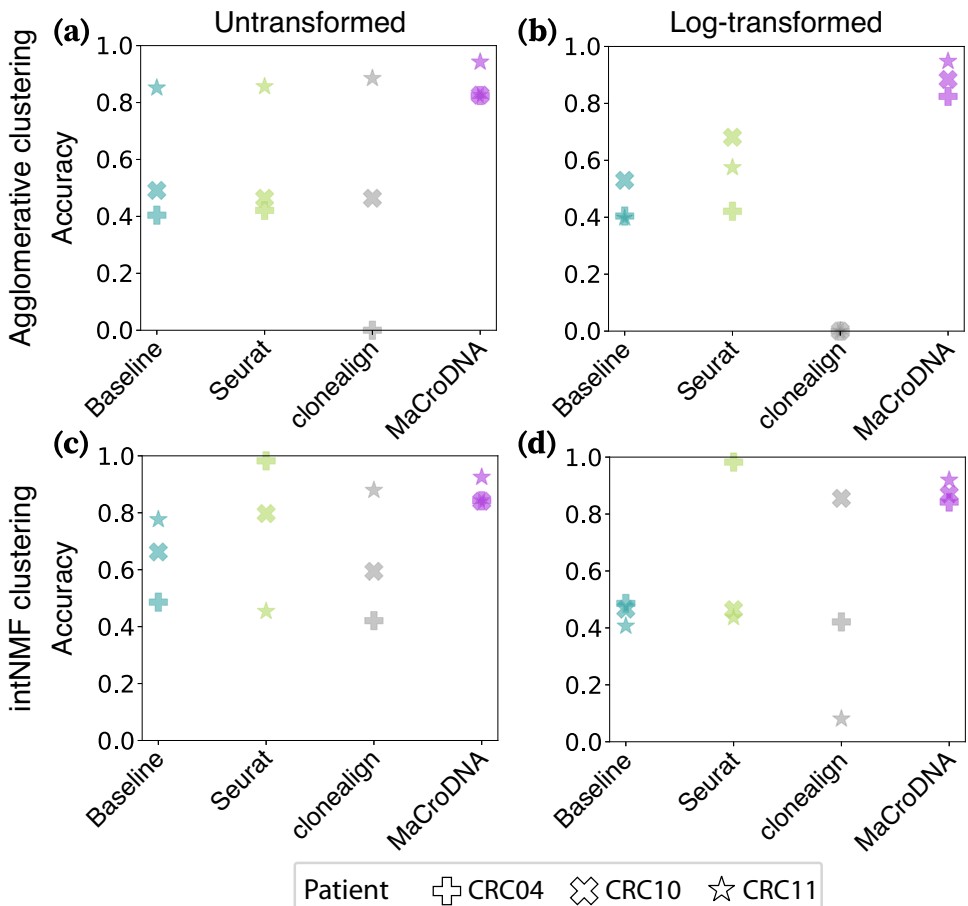

**Fig. 2 | Accuracy of the methods for the cell-to-clone assignment.** The accuracy of a method was calculated as the percentage of correct predictions (regardless of which clone a scRNA-seq cell was assigned to among all predictions made by the method in each patient's data). Each panel displays the methods' accuracy across the three patients when the clustering algorithm and the input data to the clustering algorithm were **a** the agglomerative clustering and the original data, **b** the agglomerative clustering and the log-transformed data, **c** the intNMF clustering and the original data, and **d** intNMF and the log-transformed data. The shapes of the markers correspond to different patients (see the figure legend). The colors of the markers indicate a particular method: cyan for the baseline method, green for Seurat, gray for clonealign, and purple for MaCroDNA. Source data are provided as a Source Data file.

Next, we assessed the performance of the methods in predicting the clonal prevalences. Figure 3 shows the true and predicted clonal prevalences. Each data point in Fig. 3 shows the proportion of a particular clone according to the ground truth on the *x*-axis and the proportion of the same clone predicted by a method on the *y*-axis. Here, again, we have used the results of agglomerative clustering at the highest-resolution setting, i.e., four clusters per patient (12 clusters in total). In such a plot, the less deviated a method's results are from the diagonal line, the more correlated the true and predicted clonal prevalences are. As shown in Fig. 3, MaCroDNA's results were more aligned with the diagonal line compared to Seurat's and clonealign's in all cases. The most diverged cases were observed in clonealign's results where either a relatively large clone in the ground truth was assigned almost no scRNA-seq cell or a small clone was assigned most of the scRNA-seq cells. It is worth noting that the random baseline was as accurate as MaCroDNA in all cases.

This experiment emphasized the importance of respecting the same underlying clonal abundance in the two modalities as a determining factor in the methods' performance. Moreover, the performance of the random baseline in the two above experiments showed that a method could be accurate in predicting the clonal prevalence while inaccurate in the cell-to-clone assignment.

While assuming similar clonal abundance across modalities favors MaCroDNA over existing methods, significant deviations from this assumption can impact its accuracy. We conducted a comprehensive study to determine the conditions and the extent to which discrepancies in the clonal proportions between the two modalities affect MaCroDNA's accuracy (see Section 3.1, "Investigating the effect of imbalance in clonal proportions across modalities in CRC patients" in Supplementary Information).

## Genomic heterogeneity drives differential expression of cancer-related genes in high-grade Barrett's esophagus and esophageal adenocarcinoma

To demonstrate the utility of MaCroDNA within the field of cancer biology, we applied it to previously published BE data[32]. In the original study, scDNA-seq and scRNA-seq experiments were performed on multiple biopsies to identify distinctive genomic and transcriptomic—including copy number and gene expression—signatures of NDBE, dysplastic BE, and EAC. Within individual cells, changes to gene expression can result from *cis* or *trans* genomic mutations. This, however, is just one contribution among a multitude of factors. Genetically identical cells that have differentiated into distinct cell types will exhibit completely different gene expression profiles[46]. Transcriptomes undergo massive changes as cells proceed through their cell cycles[47]. Cells also respond to transient environmental signals such as hormones or the availability of nutrients[48].

We hypothesized that within healthy tissue, the differential expression would mostly be driven by factors other than mutations, but as BE develops and progresses to dysplasia and EAC, the

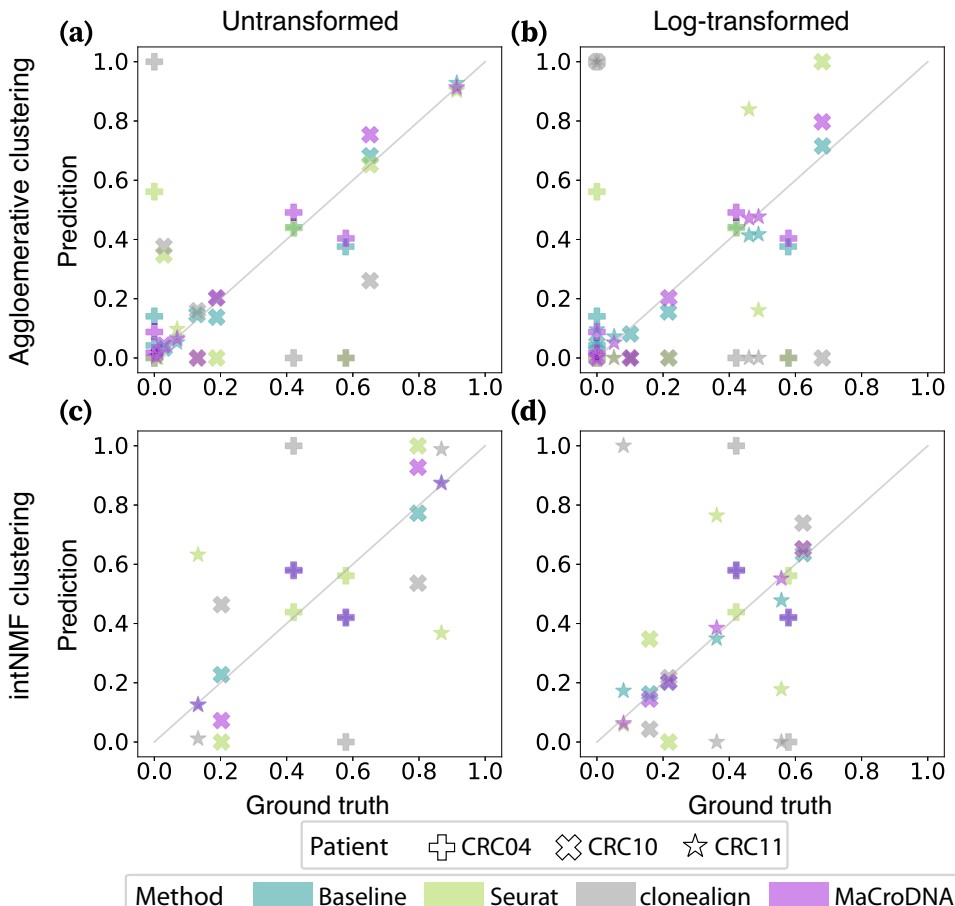

**Fig. 3 | Correlation between the true and predicted clonal prevalences from different methods.** In each plot, the *x* and *y*-axis indicate the true and predicted clonal prevalences, respectively. Each dot shows the true and predicted clonal prevalence of a patient-specific clone obtained from a method when the clustering algorithm and the input data to the clustering algorithm were **a** the agglomerative clustering and the original data, **b** the agglomerative clustering and the log-transformed data, **c** intNMF and the original data, and **d** intNMF and the log-transformed data. The shapes of the markers correspond to different patients (see the figure legend). The colors of the markers indicate a particular method: cyan for the baseline method, green for Seurat, gray for clonealign, and purple for MaCro-DNA. Source data are provided as a Source Data file.

accumulation of mutations would cause genomic mutations to become an increasing driver of transcriptomic heterogeneity. To test our hypothesis, we first selected the biopsies for which both the scDNA-seq and scRNA-seq were provided. This includes one EAC biopsy, eight BE biopsies, and one biopsy each from healthy esophageal (ESO) and gastric cardia (CARD) tissues. After filtering and normalizing DNA and RNA datasets, we created separate DNA and RNA read count tables for genes with mapped RNA reads, which also overlapped bins with mapped DNA reads (Table 2). We then used MaCroDNA to find the correspondence between the scRNA-seq and scDNA-seq cells in the two datasets. We then treated each matched pair of RNA-DNA cells identified by MaCroDNA as a single cell for which we have both RNA and DNA information. Since no ground truth correspondence information was available for these datasets, MaCroDNA was an indispensable part of our analysis. To obtain confidence scores for MaCroDNA's assignments, we performed random assignment tests on all BE biopsies, which demonstrated that MaCroDNA's results were statistically significant (see Section 3.2, "Random assignment test for BE biopsies" in Supplementary Information). Furthermore, our stability analyses of MaCroDNA's assignments for BE biopsies revealed that the stability and definiteness of MaCroDNA's assignments were proportional to the heterogeneity of the biopsies, with greater stability and definiteness observed in more heterogeneous (HGD and EAC) biopsies (see Sections 3.3, "Stability analysis of MaCroDNA's

assignments for BE biopsies", and 3.4, "Retrieval of BE biopsy labels" in Supplementary Information).

To measure the contribution of genomic mutations to transcriptomic changes, we used the $K$ statistic of phylogenetic signal, which quantifies correlations between a continuous trait and a phylogenetic tree[49]. For our analysis, the trait was the normalized read count for a particular gene. The phylogenetic tree, which was estimated from normalized binned DNA read counts as a proxy for copy number profiles and encoded cell lineage information. Using a phylogenetic tree to model cell lineages is natural, given the bifurcating nature shared by such trees and by the cell cycle, and correlations between cell transcriptome and lineage will result from mutations or environmental factors that are both specific to sub-lineages of cells within a biopsy and have some effect on the transcriptome. $K$ has an expected value of 1 when traits evolve along the tree following a Brownian motion (BM) model, where the variance of a trait increases linearly with time, so a $K$ of 1 would represent a very strong correlation between cell transcriptome and lineage. $K < 1$ or $K > 1$ when the correlation is weaker or stronger, respectively, than expected under BM[49].

Initial inspection of the distribution of $K$ values for all postfiltering genes within each biopsy revealed that the bulk of gene expression patterns exhibit correlations with the cell lineage phylogeny below what would be expected under BM, regardless of tissue type or BE staging (Fig. 4). Since a calculation for the expected value of

**Table 2 | Cells and genes from esophageal biopsies after post-processing and analysis**

| Biopsy | | Cell counts | | All overlapping genes | | COSMIC genes $p < 0.05$ and $K^* > 1$ | |
|---|---|---|---|---|---|---|---|
| PID[a] | Histology | RNA | DNA | No. any $p$ and $K^*$ | No. $p < 0.05$ and $K^* > 1$ | No. | Max $K^*$ (gene) |
| 20 | CARD[b] | 249 | 342 | 6671 | 98 | 7 | 1.12 (*MUC1*)[c] |
| 20 | ESO[d] | 181 | 339 | 7581 | 2 | 0 | — |
| 9 | NDBE[e] | 160 | 362 | 3783 | 35 | 4 | 1.01 (*ERBB2*) |
| 14 | NDBE | 259 | 200 | 9213 | 8 | 0 | — |
| 16 | NDBE | 237 | 227 | 7217 | 0 | 0 | — |
| 6 | LGD[f] | 221 | 353 | 7818 | 58 | 0 | — |
| 19 | LGD | 354 | 317 | 8566 | 5 | 0 | — |
| 6 | HGD[g] | 189 | 312 | 8308 | 485 | 35 | 1.54 (*ERBB2*) |
| 14 | HGD | 172 | 117 | 7282 | 2612 | 119 | 1.27 (*CXCR4*) |
| 20 | HGD[h] | 233 | 339 | 7148 | 416 | 39 | 1.33 (*PTPRC*) |
| 16 | EAC[i] | 187 | 274 | 7713 | 244 | 20 | 1.46 (*GNAS*) |

We measured the contribution of genomic mutations to transcriptomic changes by $K^*$ index of phylogenetic signal[49]. For each gene, the $p$-value was estimated by a one-sided random permutation test with 999 repetitions to test $K^*$ index against the null hypothesis of the gene expression values being randomly distributed in the phylogeny ($n$ = number of DNA cells per biopsy).
[a]Patient identification number from the original study.
[b]Healthy gastric cardia.
[c]The gene names are italicized according to the organism-specific formatting guidelines for humans.
[d]Healthy esophagus.
[e]Non-dysplastic Barrett's esophagus.
[f]Low-grade dysplastic Barrett's esophagus.
[g]High-grade dysplastic Barrett's esophagus.
[h]The first of two HGD biopsies from this patient separated by at least 5 cm.
[i]Esophageal adenocarcinoma.

$K^*$ when traits are completely independent of the phylogeny has yet to be derived, especially when considering measurement error of gene expression and estimation error of the cell lineage phylogeny, it is not possible to appraise the weakness of this correlation. Further inspection uncovered a large number of genes had expression patterns inferred as being more strongly correlated with the cell lineage phylogeny than expected under BM in HGD and EAC biopsies (Fig. 4h–k) but not in healthy tissues or less advanced stages of BE (Fig. 4a–g).

In simulation studies, a phylogenetic signal stronger than expected under BM can be induced by increasing the amount of change in a trait that occurs deeper in the tree (e.g., along internal branches) relative to the amount of change toward the tips[50]. Following that, there are complementary plausible explanations of greater-than-expected signals in the BE data, both adaptive and neutral. Sometimes large changes in gene expression may confer a selective advantage, ensuring such changes are preserved within sub-lineages of cells. For example, amplification of the HER2-encoding gene *ERBB2* results in HER2 overexpression which drives proliferation and survival, and HER2-positive cells are dependent on this overexpression to avoid growth inhibition and apoptosis[51]. While mutations that induce large changes in gene expression are rare[52], as cancer develops and progresses, mutation rates rise, increasing the chance that such mutations will occur.

After restricting our analysis to gene expression patterns with statistically significant phylogenetic signals based on the $K^*$ test[49], we identified hundreds of genes with inferred $K^* > 1$ in every HGD and EAC biopsy. In biopsies of lower-grade or healthy tissue, far fewer genes were identified following the same criteria (Table 2). Many of the identified genes are present in the COSMIC Cancer Gene Census[53] of genes causally implicated in cancer, including *ERBB2*, which had the highest inferred $K^*$ of any statistically significant COSMIC genes in two biopsies (Table 2, Supplementary Table S1).

**Computational cost**

We evaluated the computational cost of each method by measuring its runtime and peak memory usage when applied to clusters inferred by intNMF. To investigate how input size affects computational cost, we created different-sized datasets for each patient by sampling cells with replacement. For each method, we randomly sampled a proportion $\lambda$ of cells from each patient's original cell set and repeated this process 10 times. We varied $\lambda$ from 0.5 to 1 in increments of 0.1 because Seurat encountered errors when the number of cells was too small. In this way, $\lambda$ increased from 0.5 to 1 with increments of 0.1. Since the number of genes can also have a significant impact on computational cost, both `2000_genes_log` and `all_genes_log` (see "Preprocessing on CRC dataset") were used as the preprocessed data for MaCroDNA, resulting in two versions: MaCroDNA_2000 and MaCroDNA.

The simulation was run on Ubuntu 20.04.4 using one core of an Intel Core i9-10900K CPU. To measure the runtime of a method, first, we calculated the method's average runtime for each patient across 10 replications, next, we reported the sum of the average running times across all patients. As shown in Fig. 5a, b, MaCroDNA had the lowest runtime when input was small (e.g., for 418 and 501 cells). We observed that the runtime of clonealign and MaCroDNA increased with the size of data, while that of Seurat was relatively unchanged since its runtime depends mostly on the dimension of the latent space in canonical correlation analysis (which was fixed at 30) rather than the size of the input.

To estimate the runtime of MaCroDNA for larger samples outside of the observed range, we fitted a linear regression between MaCroDNA's running time measures and the input sizes for the original and log-transformed data separately (Fig. 5a, b). The fitted $R^2$ values in both cases were greater than 0.96. Based on the fitted line, we can extrapolate the runtime of MaCroDNA to be around 10 min for 20,000 cells.

For memory usage of each method, we reported its peak of memory usage across all patients and all replications per sampling rate. The peak memory usage of each method is presented in Fig. 5c, d. MaCroDNA outperformed the other methods with only 0.18 GB for 835 cells. Seurat, on the other hand, required 1.15 GB of memory for the smallest input with 418 cells, which is 6 times the highest peak memory usage of MaCroDNA. Similarly, the lowest peak memory usage of clonealign was 10 times the highest peak memory usage of MaCroDNA, with 1.8 GB for 418 cells. The above experiments showed that MaCroDNA was very efficient in both time and memory consumption compared to the other two methods.

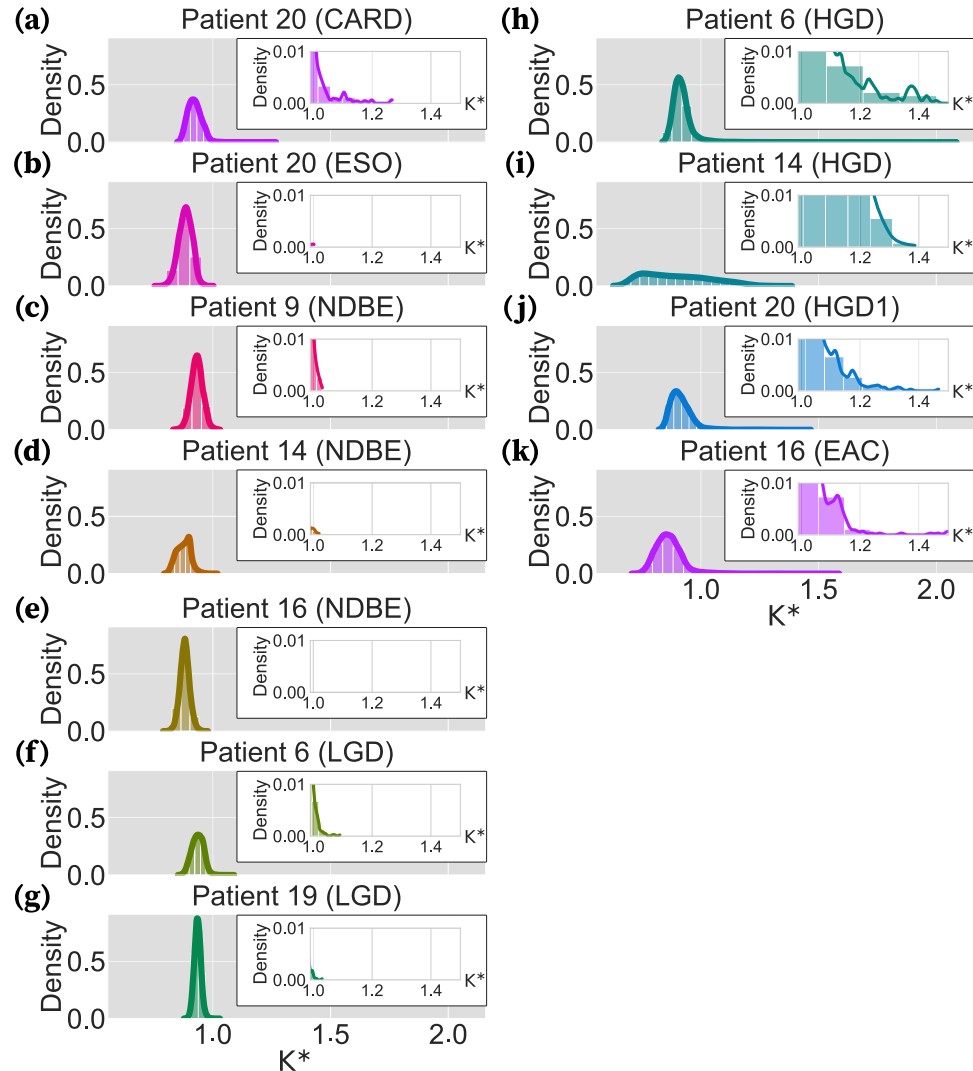

**Fig. 4 | Distribution of the _K'_ indices for all genes in each biopsy.** Each panel illustrates the distribution curve of the _K'_ indices calculated by phylosignal for all the genes in a biopsy. In each panel, the x-axis and y-axis are the _K'_ index values and their density, respectively. The left panels are the distribution curves of the healthy, NDBE, and LGD biopsies (**a**–**g**). The distribution curves of the HGD and EAC biopsies are shown in the right panels (**h**–**k**). Inserts within each panel magnify the bars and curve for $1 < K' < 1.5$. Each patient's curve is shown in a different color for better distinction. Source data are provided as a Source Data file.

## Discussion

High-throughput scRNA-seq and scDNA-seq can measure either the genomic or transcriptomic contents of the cells. Such abundant scDNA-seq and scRNA-seq datasets, if integrated precisely, would provide an unprecedented opportunity for studying the transcriptomic effects of genomic alterations. Ensuring this precise integration is not an easy task as the integration of the single-cell data across different measurements and samples—in a broader perspective —has been identified as one of the eleven grand challenges in single-cell data science[54]. In this work, we developed a method for mapping the single cells with only gene expression measurements to single cells with only copy number values from the high-throughput scRNA-seq and scDNA-seq data. Our method, named MaCroDNA, finds the correspondence between the scRNA-seq and scDNA-seq cells by maximizing the sum of the Pearson correlation coefficients between the paired cells. We also pose a restriction on this correspondence: The number of scRNA-seq cells assigned to a scDNA-seq cell should be limited as much as possible to avoid trivial solutions (see "MaCroDNA" for more details).

We assessed MaCroDNA's performance against clonealign[21]—as the state-of-the-art method for integrating the scRNA-seq and scDNA-

seq data—and Seurat[19]—as the representative of the commonly used methods in the single-cell multi-omic studies—through their application to a CRC dataset[31]. This dataset contains the cells measured for both gene expression and copy number data. We exploited such cells' information as the ground truth and used it for evaluation purposes. While simulation studies based on synthetic single-cell gene expression and copy number data can be very insightful for the evaluation of integration tools, such a task would be very challenging as our knowledge of the DNA-to-RNA process is too limited to produce realistic simulations.

We evaluated the accuracy of the methods for two inference tasks, namely the cell-to-clone assignment and prediction of the clonal prevalences. MaCroDNA outperformed clonealign and Seurat in both tasks. We hypothesize that the key to the better performance of MaCroDNA is restricting the number of the scRNA-seq cells being assigned to each scDNA-seq cell. This restriction helped MaCroDNA implicitly assume the same underlying clonal prevalences in the two modalities, whereas clonealign and Seurat do not—either explicitly or implicitly—account for such an assumption.

Among the three methods, clonealign showed the poorest performance. It is worth noticing that the only clonal inputs to clonealign

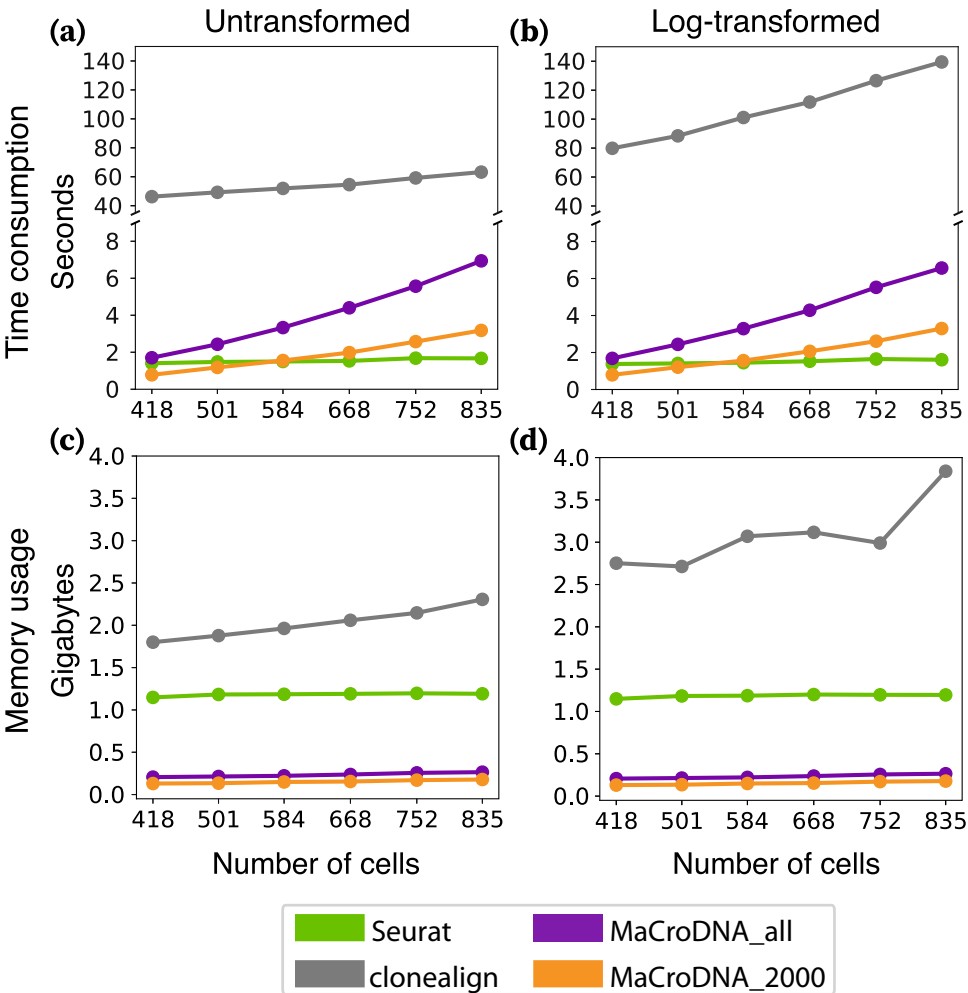

**Fig. 5 | Time consumption and memory usage of clonealign, Seurat, and MaCroDNA using different amounts of cells.** The data was sampled separately in each patient at the same sampling rate. For MaCroDNA, both `2000_genes_log` and `all_genes_log` are used as the preprocessed data. **a** is the time consumption using the untransformed clones. **b** is the time consumption using the log-transformed clones. **c** is the peak memory usage using the untransformed clones. **d** is the peak memory usage using the log-transformed clones. The different colors distinguish the results of different methods: green for Seurat, gray for clonealign, orange for MaCroDNA with `2000_genes_log` data, and purple for MaCroDNA with `all_genes_log` data. Source data are provided as a Source Data file.

are the clone-specific copy numbers (a CNA profile for each clone). Therefore, clonealign is oblivious to the distribution of the clones in the tumor. Another contributing factor that may have degraded the performance of clonealign could be its dosage function. Although MaCroDNA and clonealign are both based on the assumption that an increase (decrease) in the copy number of a gene can increase (decrease) the corresponding gene expression value at that gene, the linear dosage function of clonealign that maps the copy number of a gene to its expression is capped at an upper limit[21]. Since the Pearson correlation—which we used as our dosage function—was not restricted or capped, we speculate that limiting the linearity between copy number and gene expression values in clonealign has had a negative impact on its performance.

Additionally, we made efforts to investigate the behavior of clonealign as thoroughly as possible using the CRC dataset. We observed that in the first iteration of its optimization process, clonealign assigns all or most of the scRNA-seq cells to the clone(s) where every gene has the same copy number or the proportion of the clone's most common copy number is high. The lack of a detailed description of Evidence Lower Bound (ELBO) for the variational approximation hindered our ability to further determine the root cause of this behavior (see Supplementary Note 1).

Although our experiments for evaluation of MaCroDNA's accuracy showed the robustness of our method to the choice of clustering technique, we are interested in a simultaneous inference of the scDNA-seq clones and cell-to-clone assignments for future research. We expect this approach to result in better accuracy. Furthermore, unlike statistical methods such as clonealign, MaCroDNA does not provide probabilistic confidence measures for the assignments. As a future direction, we will explore adding uncertainty to MaCroDNA's framework.

Our re-analysis of the BE dataset demonstrates its suitability by identifying a strong phylogenetic signal of the HER2-encoding gene *ERBB2* in HGD and NDBE biopsies. This identification supports previous findings of intratumoral heterogeneity of HER2 expression in EAC but also suggests that this heterogeneity can arise in earlier stages. Those previous findings were based on in situ hybridization to assess *ERBB2* copy number amplification[55] and immunohistochemistry to assess HER2 protein overexpression[56]. Unlike in situ hybridization or immunohistochemistry, which must be tailored to individual genes in advance, MaCroDNA unlocks the entire genome and transcriptome for interrogation and further study.

To assess the performance of MaCroDNA on the BE data, in the absence of ground truth information about the correspondence

between cells, we performed comprehensive resampling and stability analyses. While MaCroDNA's results were statistically significant, we observed that the cell-to-clone assignments were less stable and definite in highly homogeneous—including healthy and NDBE—biopsies. This suggests that matching cells is distinctively challenging in homogeneous samples and highlights a biologically driven issue that may arise when utilizing MaCroDNA in such cases.

MaCroDNA is a simplistic approach—in both the assumptions and implementation—that provides accurate results, and as a reliable and easy-to-run tool, it would be a suitable method of choice for biological scientists interested in single-cell multi-omic (especially cancer) studies.

## Methods

### Description of CRC dataset

To assess the performance of the integration methods, we used a colorectal cancer (CRC) dataset introduced by Bian et al.[31]. To generate this dataset, the authors used the single-cell triple omics sequencing2 (scTrio-seq2) sequencing technique. This sequencing technology provides the joint measurement of somatic copy number variations, DNA methylation, and gene expression read counts from the same cell. In the original study, for six patients, namely CRC01, CRC02, CRC04, CRC09, CRC10, and CRC11, the joint copy number and gene expression measurements were provided. Note that not all the cells in these patients have both scRNA-seq and scDNA-seq data. The sequenced cells with joint copy numbers and gene expression values from these patients are valuable for evaluation purposes. However, we did not include all patients' datasets in our analysis.

The authors used Tang protocol[57] for transcriptome sequencing library preparation of CRC01 and CRC02, while for the other patients, multiplexed scRNA-seq method[58] was used. To maintain consistency across our experiments, we did not consider CRC01 and CRC02 in our analysis. Besides, the data of the patient CRC09 contained only 13 cells with both scRNA-seq and scDNA-seq data, which was insufficient for a statistical method such as clonealign. Finally, only the data from patients CRC04, CRC10, and CRC11 were used to evaluate MaCroDNA and the other integration methods.

### Preprocessing on CRC dataset

First, we filtered out the non-informative genes from the scRNA-seq gene expression data by keeping only genes that were expressed as non-zero in more than 1% of cells. The genes obtained from filtering are used for further preprocessing in the integration methods and the clustering techniques. In the following, we will describe our data preparation procedures for the integration methods and the clustering methods. Specifically, data preparation for each of the existing methods is based on the instructions in their original studies.

clonealign requires removing the X-chromosome genes prior to the analysis since the presence of the X-chromosome in the data will violate the method's assumption about the relationship between the copy number and gene expression values. Additionally, as recommended in the original clonealign study, genes with zero variance in copy number for each patient and those with zero copy number in at least one cell were also removed. In addition to removing genes with zero variance, we assessed the impact of adding a pseudocount value of 1 to all absolute copy number values before inputting data into clonealign and then compared the two approaches (see Section 2.5, "Investigating clonealign's accuracy by adding pseudocount values to the inputs" in Supplementary Information).

Seurat requires different preprocessing procedures. We followed the same procedures in the original paper. First, all cells were log-transformed using a size factor of 10,000 molecules for each cell. Then, z-score normalization was applied to the log-transformed data. Next, a feature selection was done to scRNA-seq data and scDNA-seq data separately, using `FindVariableFeatures` function in Seurat v3.

Following the feature selection on the single measurements, feature selection for integrated analysis was applied to scRNA-seq data and scDNA-seq data together using `SelectIntegrationFeatures` function in Seurat v3. Finally, the top 2000 genes were selected.

In addition to Seurat's and clonealign's preprocessing procedures, we considered two other preprocessing strategies for MaCroDNA, resulting in four strategies in total. The first strategy is Seurat's preprocessing, named as `2000_genes_log` in the figure legends in the Supplementary Information. This includes log transformation of the data, z-score normalization with 10,000-factor size, feature selection, and the selection of the top 2000 genes. All these steps were applied sequentially to the input. The second one includes the log transformation and z-score normalization (with 10,000 factor size), but all genes were kept. This preprocessing is named as `all_genes_log` in the supplementary figures. The third one is the clonealign's preprocessing, tagged with `noX_genes_raw`. Finally, the last strategy is `all_genes_raw`, where all genes were used, and the data were kept in the original scale. In Figs. 2 and 3, we used `2000_genes_log` preprocessing for Seurat (as it is recommended by the authors) and `all_genes_log` for MaCroDNA. For clonealign, we applied `noX_genes_raw`. The full comparison of these preprocessing strategies and their effect on the accuracy of the methods can be found in Supplementary Figs. S1–S18.

Apart from data preprocessing for the integration methods analyses, the CNA data used for obtaining clusters was also preprocessed. Here, we use two strategies: clustering on the original CNA data and clustering on the log-transformed CNAs. The CNA data can have a right-skewed distribution. Thus, $\log(x+1)$ was applied, where $x$ is the original copy number. Through all experiments, both data scaling strategies were used for clustering and clonal inference.

### Preprocessing on BE dataset

For scRNA-seq data of each biopsy, first, we filtered out the single cells with less than 3000 transcripts from the data, and next, we kept the genes that were expressed by at least three transcripts in at least one cell, according to the filtering procedure in ref. 32. For numerical stability of our calculations, we added a pseudocount of 1 to all entries of the scRNA-seq count tables which is a common practice in gene expression analysis[59]. To circumvent the effect of unequal sequencing depth, we performed RPM (reads per million mapped reads) normalization by first dividing each gene's expression value by the total number of reads in the cell and then scaling this ratio by one million. Finally, we log-normalized the gene expression values by applying $\log(x+1)$ where $x$ is the gene expression value and $\log(\cdot)$ is the natural logarithm.

The scDNA-seq data was provided to us as the binned genomes containing the number of mapped reads in each copy number bin. Similar to scRNA-seq data, we filtered out the scDNA-seq cells with less than 3000 mapped reads. Next, we added the pseudocount of 1 to the number of mapped reads in each bin. Following the instructions from the original study[32], we divided the copy number bin counts by their median and then multiplied by 2 to obtain a rough estimation of the copy number values. In the next step, we log-normalized the copy number values using $\log(x+1)$.

Running MaCroDNA on the scDNA-seq and scRNA-seq data requires the count tables from the two modality with the same set of genes. As mentioned above, the given data was the binned copy number data. Therefore, we first annotated the copy number bins by searching for the genes whose genomic coordinates lie within the bins. We downloaded the GENCODE[60] annotation file for GRCh37 assembly, Human Release 19 (GRCh37.p13), and extracted the names and coordinates of all known protein-coding genes. Next, for each copy number bin, we annotated it with all the genes whose starting and ending points were within the bin. For each gene, we considered the copy number of the corresponding bin as that gene's copy number value.

Lastly, for each biopsy, we found the genes shared between the scRNA-seq and scDNA-seq datasets and performed MaCroDNA on this set of genes.

## Clonal inference by intNMF

intNMF is a clustering method based on non-negative matrix factorization[29,30,61]. intNMF is able to identify the optimal number of clusters for each data type in a multi-omic dataset. Its clustering algorithm can also be applied to datasets with single data types (such as scDNA-seq data in CRC dataset). In intNMF, the cluster prediction index (CPI) can evaluate the goodness of a particular number of clusters by measuring the adjusted rand indices[62] among the clustering assignments (for more details, see ref. 40). The optimal number of clusters is the one that yields the maximum CPI value.

We applied intNMF to the scDNA-seq cells of each patient to infer the clusters. First, we ran the function `nmf.opt.k` from the R package of intNMF v1.2.0 to identify the optimal number of clusters in each patient. Next, to perform clustering on the scDNA-seq cells and acquire the clustering membership assignments, we ran the function `nmf.mnnals` with the optimal number of clusters obtained from the previous step as the input. Given the original data, intNMF detected two clusters in each of the patients, CRC04, CRC10 and CRC11. These patient-specific clusters are denoted by crc04_clone0, crc04_clone1, crc10_clone0, crc10_clone1, crc11_clone0, and crc11_clone1 in Supplementary Fig. S16. Using the log-transformed data as input, intNMF identified two clusters in patient CRC04 denoted by crc04_clone0 and crc04_clone1 in Supplementary Fig. S18. Three clusters were identified in each of the patients, CRC10 and CRC11. These clusters are denoted by crc10_clone0, crc10_clone1, crc10_clone2, crc11_clone0, crc11_clone1, and crc11_clone2, in Supplementary Fig. S18.

## Clonal inference by agglomerative clustering

In addition to intNMF, we applied agglomerative clustering to obtain scDNA-seq clones from each patient's cells. The Python implementation of agglomerative clustering provided in sklearn package v0.23.2 was directly applied for obtaining 4 clusters in each patient. We used Euclidean distance as the distance metric and average as the linkage criterion. By obtaining 4 clusters for each patient, we inferred 12 clusters in total across all patients. Since we sought to investigate the effect of varying clustering resolution on the methods' accuracy, we applied the following procedure to decrement the number of clusters:

1. For each patient having more than two clusters, calculate all pairwise distances between the clusters within the patient. The distance between two clusters is the Euclidean distance between the averaged copy number profiles representing the clusters.
2. Among all pairwise distance values, identify the pair of clusters (within the same patient) with the minimum distance and merge them into a single cluster.
3. Repeat steps 1 and 2 until only two clusters remain for each patient.

This way, we varied the resolution of the clusters obtained from all patients, ranging from 6 clusters to 12 clusters in total. The patient-specific clones for different resolutions are tagged in Supplementary Figs. S2–S7 and S9–S14.

## Random baseline

To evaluate the performance of MaCroDNA and other methods, we established a random baseline. In this random baseline, each scRNA-seq cell is assigned to a scDNA-seq cell. This scDNA-seq cell is selected by random sampling from all scDNA-seq cells in the same patient with replacement. Following this random method for assignment, we can calculate for each patient $i$, the expected number of scRNA-seq cells

(whose true clone id is $c_{\text{true}}$ according to the ground truth information and the current clustering assignment) that are being assigned to the scDNA-seq cells with clone id $c_{\text{predicted}}$. Let $n_{\text{true,predicted}}$ denote the above-expected number. Then we have:

$$n_{\text{true,predicted}} = n_{c_{\text{true}}}^{\text{RNA}} \cdot n_{c_{\text{predicted}}}^{\text{DNA}} / N_i^{\text{DNA}}, \tag{1}$$

where $n_{c_{\text{true}}}^{\text{RNA}}$ is the number of scRNA-seq cells that belong to clone $c_{\text{true}}$ in patient $i$, $n_{c_{\text{predicted}}}^{\text{DNA}}$ is the number of scDNA-seq cells in clone $c_{\text{predicted}}$ in patient $i$, and $N_i^{\text{DNA}}$ is the total number of scDNA-seq cells in patient $i$. This expected number gives us a measurement for the accuracy of the baseline method.

## Phylogenetic signal analysis on the BE dataset

To prepare the inputs for phylogenetic signal analysis, we first inferred the single-cell CNA phylogenetic tree of each biopsy using the UPGMA method. We computed the Euclidean distances between all pairs of the cells and passed the pairwise distance matrix as the input to `upgma` function in phangorn R package v2.11.1[63] to infer the UPGMA tree. As mentioned earlier, we used the copy number values of the genes shared between the two modalities to perform MaCroDNA. Here, however, for phylogenetic inference, we used the complete set of copy number bin counts in order to take advantage of all the available evolutionary information from scDNA-seq data for the phylogenetic tree reconstruction.

In addition to the phylogenetic tree, phylosignal requires a table of traits for each cell. Given the cell-to-cell assignments inferred by MaCroDNA, for each scDNA-seq cell, we provided the vector of gene expression values of the paired scRNA-seq cell as the cell's traits to phylosignal. Of note, some of the scDNA-seq cells were assigned to multiple scRNA-seq cells by MaCroDNA. In such cases, for one particular scDNA-seq cell, we ranked the assigned scRNA-seq cells according to the iteration number they were assigned to the scDNA-seq cell (see "MaCroDNA for $N_G > N_C$" for more details). An earlier iteration, in this case, means a better correspondence score. Therefore, we selected the scRNA-seq cell that was assigned to the scDNA-seq in the earliest iteration. Here, for each paired scRNA-seq, we passed the gene expression of all genes (rather than only the shared genes with the DNA data) as the traits to phylosignal.

We ran the `phyloSignal` function in phylosignal's R package v1.3 with the UPGMA tree and the single-cell gene expression as inputs to compute the $K^*$ statistic for each gene. That function computes the $p$-value of the $K^*$ index by testing it against the null hypothesis of the gene expression values being randomly distributed in the phylogeny[64]. Throughout our analysis of the BE dataset, we considered the indices with $p$-values less than 0.05 as the significant indices.

The $K^*$ statistic[49], as a measure of phylogenetic signal, is calculated for each gene using the equation

$$K^* = \left(\frac{\text{MSE}^*}{\text{MSE}}\right) \bigg/ \mathbb{E}_{\text{BM}}\left[\frac{\text{MSE}^*}{\text{MSE}}\right] \tag{2}$$

where $\text{MSE}^*$ is the mean squared error between the observed gene expression values at the tips of the phylogenetic tree and their mean. This is calculated as:

$$\text{MSE}^* = \frac{(\mathbf{X} - \bar{a})^{\top}(\mathbf{X} - \bar{a})}{(n-1)} \tag{3}$$

where $\mathbf{X}$ is the vector of observed gene expression values of the scRNA-seq cells for a gene, $\bar{a}$ is the average gene expression value, and $n$ is the number of scRNA-seq cells[49]. MSE is the mean squared error between the gene expression values of cells and their phylogenetically correct

mean[49], calculated using the equation

$$MSE = \frac{(\mathbf{U} - \hat{a})^{\top}(\mathbf{U} - \hat{a})}{n - 1} \tag{4}$$

where $\mathbf{U} = \mathbf{DX}$ is the transformed $\mathbf{X}$ obtained from generalized least-squares[49]. The connection to the phylogenetic structure is made through matrix $\mathbf{D}$, which satisfies the constraint $\mathbf{DVD}^{\top} = \mathbf{I}$ where $\mathbf{V}$ is the phylogenetic variance-covariance matrix[49,65,66] and $\mathbf{I}$ is the identity matrix. The phylogenetic variance-covariance matrix is calculated based on the branch lengths of the phylogeny: the variance of each cell is defined as the sum of the branch lengths starting from the root to the cell, and the covariance between two cells is the sum of the branch lengths from the root to their most recent common ancestor. $\mathbb{E}_{BM}$ in equation (4) indicates the expected value of the MSE*/MSE ratio under BM[49].

## MaCroDNA

The input of MaCroDNA consists of two single-cell datasets supposedly sampled from the same tissue: one contains absolute copy numbers (or their log-transformations) of some single cells at a particular set of genes. The other dataset consists of gene expression read counts (or their log-transformed values) belonging to another set of single cells at the same genes as in the CNA dataset.

These two datasets are provided as inputs to MaCroDNA in the form of matrices $\mathbf{C}$ and $\mathbf{G}$ for CNA and gene expression data, respectively. Formally, let $\mathbf{C} = (c_{ij})_{1 \le i \le N_C, 1 \le j \le M}$ be the matrix of single cells' CNA information. Here, $N_C$ represents the number of single cells, $M$ denotes the number of genes, and $c_{ij}$ is the absolute copy number or its log-transformed value in the $i^{th}$ single cell at the $j^{th}$ gene. Similarly, $\mathbf{G} = (g_{ij})_{1 \le i \le N_G, 1 \le j \le M}$ is the matrix of single cells' gene expression information, where $N_G$ denotes the number of single cells, $M$ denotes the number of genes, and $g_{ij}$ is the gene expression read count or its log-transformed value in the $i^{th}$ single cell at the $j^{th}$ gene. Each row in $\mathbf{C}$ or $\mathbf{G}$ represents the CNA profile or gene expression vector of a single cell. We use $\mathbf{c}_i$ and $\mathbf{g}_i$ to denote the $i^{th}$ cell's CNA profile and gene expression vector, respectively.

Given the above matrices, MaCroDNA associates each cell in gene expression data (scRNA-seq cell) to exactly one cell in CNA data (scDNA-seq cell) according to a criterion described in the following. This correspondence is presented in the form of a binary matrix, $\mathbf{I} = (I_{ij}) \in \{0,1\}^{N_G \times N_C}$, where $I_{ij}$ is 1 if the $i^{th}$ scRNA-seq cell is associated with the $j^{th}$ scDNA-seq cell, otherwise, $I_{ij}$ is 0. Of note, since each scRNA-seq cell has exactly one corresponding scDNA-seq cell, each row of the correspondence matrix has exactly one entry of value 1.

In the following, we discuss how MaCroDNA infers the above-mentioned correspondence matrix by, first, describing our polynomially solvable algorithm for cases where the number of scRNA-seq cells is less than or equal to the number of scDNA-seq cells ($N_G \le N_C$) and second, our heuristic for datasets where the number of scRNA-seq cells is more than the number of scDNA-seq cells ($N_G > N_C$). Finally, we discuss how our solution for the former constitutes the basis for the solution to the latter.

**MaCroDNA for $N_G \le N_C$.** As mentioned above, we aim to associate each scRNA-seq cell with exactly one scDNA-seq cell. When $N_G \le N_C$, we add another constraint to this problem: each scDNA-seq cell is associated with at most one scRNA-seq cell. The main reason for this constraint is to prevent the situation where a significant number of scRNA-seq cells are associated with a particular scDNA-seq cell—which violates our underlying assumption of the same clonal distribution across the two modalities. An alternative to this constraint is to consider an upper limit on the number of correspondents for each scDNA-seq cell. Pre-specifying or inferring this upper limit for each scDNA-seq cell, however, would be a challenging task that we leave for future exploration.

Considering the above conditions, we require the solution to be of the highest similarity score between the corresponding pairs. We formulated this optimization problem as an instance of Mixed Integer Linear Programming (MILP). Specifically, we introduce binary indicator variables $I_{ij} \in \{0, 1\}$, which is 1 when the $i^{th}$ scRNA-seq cell is assigned to the $j^{th}$ scDNA-seq cell; otherwise, $I_{ij}$ is 0. The objective function to maximize is the sum of the Pearson correlation coefficients (see Section "Calculation of correlation coefficients" for more details) of the selected pairs,

$$\max_{I_{ij}} \left\{ \sum_{i=1}^{N_G} \sum_{j=1}^{N_C} \omega_{ij} I_{ij} \right\}, \tag{5}$$

where $\omega_{ij}$ is the Pearson correlation coefficient between the $i^{th}$ scRNA-seq cell and the $j^{th}$ scDNA-seq cell. Since each scDNA-seq cell is allowed to be assigned to at most one scRNA-seq cell, at most one of the indicator variables of each scDNA-seq cell can be 1. The following inequality forces this constraint for each scDNA-seq cell:

$$\sum_{i=1}^{N_G} I_{ij} \le 1, \quad \forall j \in \{1 \le j \le N_C | j \in \mathbb{N}_0\}. \tag{6}$$

Another constraint we need to respect is that each scRNA-seq cell must have exactly one pair in scDNA-seq data. Although the constraint $\sum_{j=1}^{N_C} I_{ij} = 1$ for each scRNA-seq cell can satisfy this condition when $N_G \le N_C$, we propose two alternative constraints that yield the same results and can also be used in the cases where $N_G > N_C$ as well (see Section "MaCroDNA for $N_G > N_C$" for more details). First, we force each scRNA-seq cell to have at most one pair in scDNA-seq data by the following inequality:

$$\sum_{j=1}^{N_C} I_{ij} \le 1, \quad \forall i \in \{1 \le i \le N_G | i \in \mathbb{N}_0\}, \tag{7}$$

and then, we restrict the total number of assignments to be exactly equal to the number of scRNA-seq cells by the following equality,

$$\sum_{i=1}^{N_G} \sum_{j=1}^{N_C} I_{ij} = N_G. \tag{8}$$

This is particularly useful when we need to select a certain number of assignments from all the possible choices in our heuristic algorithm. Thus, the overall optimization problem will be

$$\max_{I_{ij}} \left\{ \sum_{i=1}^{N_G} \sum_{j=1}^{N_C} \omega_{ij} I_{ij} \right\},$$

$$\text{subject to } \sum_{i=1}^{N_G} I_{ij} \le 1, \forall j, \quad \sum_{j=1}^{N_C} I_{ij} \le 1, \forall i, \quad \sum_{i=1}^{N_G} \sum_{j=1}^{N_C} I_{ij} = N_G, \tag{9}$$

which can be solved using well-known MILP-solvers such as Gurobi (see https://www.gurobi.com) and CPLEX (see https://www.ibm.com/analytics/cplex-optimizer). After solving the optimization problem in equation (9), each indicator variable will be an entry of the binary correspondence matrix, $\mathbf{I} = (I_{ij})$.

It is worth mentioning that this problem can be viewed as an instance of maximum weight perfect matching on a complete bipartite graph, also known as assignment problem. To reformulate as such, we need to introduce a graph $\mathcal{G} = (V, E)$ consisting of $V$ vertices and $E$ edges. Here, the vertices represent single cells from scDNA-seq and scRNA-seq. All possible pairs of single cells in scRNA-seq and scDNA-seq are connected by edges that are weighted with the Pearson correlation coefficient between the pair. The goal of the maximum weight

perfect matching problem is to find a one-to-one matching between the nodes in the bipartitions such that the sum of the edge weights connecting the pairs is maximized. The well-known Hungarian algorithm[67] can solve this problem in polynomial time[39]. The time complexity of this algorithm is $\mathcal{O}(|V|^3)$ where $V$ is the set of nodes in the bipartite graph[68,69].

The Hungarian algorithm requires the number of scRNA-seq and scDNA-seq cells to be equal which is not the case necessarily in real applications. However, we can add dummy scRNA-seq cells that are connected to scDNA-seq cells with edges weighted by negative values with large magnitudes. By doing so, the number of cells in both bipartitions will be the same and the Hungarian algorithm is applicable.

**MaCroDNA for $N_G > N_C$.** When the number of scRNA-seq cells exceeds the number of scDNA-seq cells, the situation where multiple scRNA-seq cells are assigned to a small group of scDNA-seq cells seems inevitable. Here, we propose a heuristic algorithm to regularize this assignment. In our scheme, we infer the correspondence of all scRNA-seq cells in multiple steps. Formally, our iterative algorithm consists of the following steps:

- **Initialization**: let $R^{(0)}$ be the set of scRNA-seq cells at the initialization step of the algorithm, and $N_{R^{(0)}}$ be the number of cells in set $R^{(0)}$. Note that at the initialization step, all the scRNA-seq cells are involved, so $N_{R^{(0)}} = N_G$. Perform the optimization described in the previous section in equation (9) with a slight modification to the equality constraint in equation (8): instead of $N_G$ assignments, we need to infer the best $N_{min} = \min\{N_G, N_{R^{(0)}}\}$ assignments (although in the first step, $N_{min} = N_G$, we use this general notation for the next iterations). So, the equality constraint of equation (8) becomes:

$$\sum_{i=1}^{N_G} \sum_{j=1}^{N_C} I_{ij} = N_{min}. \tag{10}$$

After inference of the best $N_{min}$ assignments, remove the scRNA-seq cells whose corresponding scDNA-seq cells are inferred from the set $R^{(0)}$. Name the new set, $R^{(1)}$, and proceed to the next step.

- **Iteration**: let $R^{(k)}$ denote the set of scRNA-seq cells at iteration $k$. Perform the optimization in equation (9) with the constraint described in equation (10). Remove the scRNA-seq cells whose assignments are determined from $R^{(k)}$. Name the new set $R^{(k+1)}$. If this set is empty, go to the termination step. Otherwise, proceed to the iteration $k + 1$.

- **Termination**: construct the binary correspondence matrix, **A**, by collecting all the assignments inferred through all the iterations. Notice that since we remove most $N_C$ cells from the set of scRNA-seq cells at each iteration, the total number of iterations will be $\lceil \frac{N_G}{N_C} \rceil$.

**Calculation of correlation coefficients.** To calculate Pearson correlation coefficients between each RNA-DNA pair of cells, we use the raw read counts in the CNA profile and gene expression vector of the cells ($\mathbf{g}_i$, and $\mathbf{c}_j$, respectively). Given two cells' vectors, $\mathbf{c}_i$ and $\mathbf{g}_j$, the Pearson correlation between them is calculated using the following formula

$$\omega_{ij} = \rho_{\mathbf{c}_i, \mathbf{g}_j} = \frac{\sum_{k=1}^{N}(c_{ik} - \mu_{\mathbf{c}_i})(g_{jk} - \mu_{\mathbf{g}_j})}{\sigma_{\mathbf{c}_i}\sigma_{\mathbf{g}_j}}, \tag{11}$$

where $\rho_{\mathbf{c}_i, \mathbf{g}_j}$ denotes the Pearson correlation between $\mathbf{c}_i$ and $\mathbf{g}_j$; $c_{ik}$ and $g_{jk}$ are $k^{th}$ elements in vectors $\mathbf{c}_i$ and $\mathbf{g}_j$, respectively.

**Reporting summary**
Further information on research design is available in the Nature Portfolio Reporting Summary linked to this article.

## Data availability
The CRC data from Bian et al.[31] is openly available in NCBI Gene Expression Omnibus (GEO) under accession number GSE97693. The Barrett's esophagus dataset from Busslinger et al.[32] is available in the European Genome-Phenome Archive (EGA) under accession number EGAS00001005221. Access to this data is controlled by a Data Access Committee. RNA and DNA read counts for both the CRC and BE were obtained directly from the authors of the original studies[31,32]. The GENCODE GFF3 annotation file for GRCh37 assembly was downloaded from https://www.gencodegenes.org/human/release_19.html. The list of cancer-related genes used in this study was downloaded from the COSMIC Cancer Gene Census web page at https://cancer.sanger.ac.uk/census. The data associated with the figures presented in this study are provided in the Source Data file. Source data are provided with this paper.

## Code availability
All codes, including the Python implementation of MaCroDNA and the scripts for data analysis in this study, are publicly available on GitHub (https://github.com/NakhlehLab/MaCroDNA) and archived on Zenodo (https://doi.org/10.5281/zenodo.10115041)[70].

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

## Acknowledgements
We thank Ana C. Afonso Silva for her comments on the implications of strong phylogenetic signals. Our research was supported in part by the National Science Foundation, grants IIS-1812822 and IIS-2106837 (L.N.).

## Author contributions
M.E., X.H., H.A.O. and L.N. designed the study. L.N. supervised the study and acquired funding and resources. All authors wrote and approved the manuscript.

## Competing interests
The authors declare no competing interests.
