## [Peer Review File · Nature Communications]

REVIEWER COMMENTS

Reviewer #1 (Remarks to the Author): Expert in single-cell multi-omics, bioinformatics, cancer genomics and evolution

This manuscript introduces MaCroDNA, a computational method to map unpaired scRNA-seq and scDNA-seq data. The method performs a simple correlation-based association between cells from both technologies. The authors apply it to one data set containing both measurements from the same cells for validation and to another with unpaired data for exploratory analysis.

MaCroDNA finds the mapping between cells from scRNA-seq and cells from scDNA-seq that maximises the sum of the Pearson correlations between each cell's gene expression and its copy number profile. This is a simple approach that is akin to the baseline method used in the simulations for the clonealign [1] paper, and which is outperformed by clonealign in that work. However, in the present manuscript, clonealign performs poorly and MaCroDNA performs very well in the real data set with ground truth. It is unclear why clonealign performs so poorly, completely failing in some cases where MaCroDNA obtains a high accuracy. Additionally, it is unclear whether the results in this ground truth data set apply to unpaired data, which is obtained by different protocols and usually yields a much larger number of cells from both modalities. For users to decide which method to use, the authors must explore the differences between the methods that explain their performance discrepancies, and ideally apply them also in simulated data.

In summary, the authors introduce a simple method that requires more careful validation and comparison with the state-of-the-art. The manuscript is well-written. We detail our concerns below.

Major comments

Our main concern is the comparison with clonealign, which the authors indicate as the state-of-the-art in unpaired scRNA-seq/scDNA-seq data mapping. MaCroDNA consistently outperforms clonealign in 3 real data sets and with multiple levels of resolution of the scDNA-seq data. All 3 data sets have at least 2 very distinct copy number clones (Fig. S5 of [2]), which makes it hard to understand why one data set in Fig S2 with agglomerative clustering yields a 0% accuracy in clonealign, and with intNMF all the data sets lead to that performance (Fig S2 c).

The results become even more puzzling when the performance of clonealign improves if the input copy numbers are log-transformed, even though the clonealign model assumes actual copy numbers as input.

The authors must elaborate on the results presented in Figure S2, in which clonealign consistently gets 0% accuracy on the CRC04 patient regardless of the number of clones. This figure is puzzling because clonealign assigns 57 cells (i.e. all cells) to CRC04_clone1 that actually belong to CRC04_clone0 in the “Reference” (which we assume means the ground truth). But if all cells in the reference belong to only one cluster, why are there two clusters being given to clonealign? What is CRC04_clone1?

Taken together, these results warrant more validation. This can include exploring what clonealign assignments are incorrect or performing simulation studies that try to recover situations in which MaCroDNA and clonealign behave more comparably, as they in principle should. Indeed, as clonealign is state-of-the-art, reporting such extremely poor performance – contradicting the original publication – asks for a closer look at the results.

Minor comments

When explaining the agglomerative clustering step to define the number of clones, the authors confusingly describe a procedure that starts by inferring 4 clones per patient independently and then merging the total of 12 clusters across the 3 patients until 2 clusters per patient are obtained. It is confusing that they describe the merging step starting from all 12 clusters across patients, instead of doing it per patient. Please re-write for clarity.

Relatedly, the figures in the supplement with confusion matrices makes it seem as if the authors ran each method on all 3 patients simultaneously. This would probably hurt Seurat’s and clonealign’s performance due to the resulting batch effects.

Please label the dots in each panel by the data set (patient) of origin.

In their clonealign script on Github, they do `dna_data[dna_data == 0] <- 1`` to avoid NaNs. But, if dna_data contains CNVs, they should set it to a very low number instead of 1. Setting CNVs of 0 to 1 may have a strong effect on the resulting assignments.

From the data on Github, we can see that some genes have huge copy numbers across all cells (e.g. 30). If the average CNV profile across cells in the same cluster is taken as clonal CNVs, and these cells are in the same cluster as cells with a much lower ploidy, it will be difficult to perform a correct assignment because the average will not represent any cell very well.

The authors mention and cite another method to perform scRNA/scDNA integration, CCNMF. They might also want to mention SCATrEx [3], which has the same purpose (with additional features).

In Section 3.8, why is the gene expression counts matrix G defined in the set of integer numbers? Don't the authors use either raw counts (positive integers) or log-normalized counts (reals)?

In Section 3.8.1, what is the difference between A and I ?

Throughout the text, it is unclear whether the input data from the scDNA modality is read counts or copy number values obtained from some other method. In Section 1, they first say that the inputs to MaCroDNA include "scDNA-seq copy number values", but then say that "these measurements are provided (...) in the form of counts matrices of single cells from the same tissue". In Section 3.8.3, they state that they use the "raw read counts in the CNA profile". Please clarify this aspect.

"We then used MaCroDNA to assemble a set of virtual cells; each virtual cell merged one cell from the scDNA-seq data set, and one cell from the scRNA-seq data set, based on the one-to-one correspondences identified by MaCroDNA." – Please clarify what a virtual cell is and how MaCroDNA obtains them.

The colors in Figure 5 are difficult to distinguish.

What is the reason behind the unusual scale in the numbers of cells in Figure 5?

References

[1] Campbell, Kieran R., et al. "clonealign: statistical integration of independent single-cell RNA and DNA sequencing data from human cancers." *Genome biology* 20.1 (2019): 1-12.

[2] Bian, Shuhui, et al. "Single-cell multiomics sequencing and analyses of human colorectal cancer." *Science* 362.6418 (2018): 1060-1063.

[3] Ferreira, Pedro F., et al. "Mapping single-cell transcriptomes to copy number evolutionary trees." *Research in Computational Molecular Biology: 26th Annual International Conference, RECOMB 2022, San Diego, CA, USA, May 22–25, 2022, Proceedings*. Cham: Springer International Publishing, 2022.

Reviewer #2 (Remarks to the Author): Expert in gastroesophageal cancer genomics, Barrett's oesophagus genomics, and single-cell genomics

The manuscript presented by Edrisi and colleagues aims to implement a simple method for the integration of single-cell RNA and DNA (defined as copy number alteration profiles) data. To achieve this goal, the authors developed MaCroDNA. This technique uses Pearson correlation to identify a

relationship between expression counts (scRNA-seq) and CNV data obtained using scDNA-seq method. The method works in the space of gene-by-cell matrix. It also aims to identify at most one cell with scRNA-seq data that correspond to one scDNA-seq cell. The authors used two publicly available datasets from single cell studies of colorectal cancer and Barrett's esophagus. Only colorectal cancer study had matched DNA and RNA data. They benchmarked their method against other tools, and they show that they tend to outperform these tools in accuracy.

The manuscript is well presented with a clear message and well-described statistical methods. The available code is well described. As the author mentioned in the discussion, their approach is "simplistic", as a result, the tool might be missing important functionality. The following question stem from the fact that cells might be misassigned in specific conditions and I would suggest that assignment of confidence scores to the matches should be a priority.

1. Firstly, the key assumption of this tool is that the scRNA-seq and scDNA-seq datasets originate from the same populations and the distribution of cell types should be similar across the modalities. However, single-cell approaches are intrinsically noisy methods where sample preparation is a key component for good recovery of cell type diversity (e.g. PMID: 32487174). As a result, the assumption that all cell types and their proportion across modalities will most likely not hold true in most settings. Could the authors elaborate on cell type assignment when there is an in-balance in cell populations across modalities? For example, they could bootstrap colorectal data by removing scDNA-seq from the cells with match RNA and DNA data to test if the RNA component is still assigned to the correct cluster.

2. Similarly, the method aims to assign only one cell from the scRNA-seq data set to exactly one cell from the scDNA-seq cell. Also, only one scRNA-seq can be assigned to each scDNA-seq cell. Could the author elaborate on how would the method behave in the following conditions that can arise due to sample preparation discrepancies: RNA: 10 cells in population A and 90 cells in population B, DNA: 90 cells in population A and 10 cells in population B. Surely, since the method aims to perform 1-to-1 matching, many cells will be assigned to wrong populations?

3. In the BE dataset, the authors used NDBE and normal samples that have few copy number changes. However, as the original study shows, there are many cell populations that can be identified using scRNA-seq. Could the authors elaborate on the stability of data integration when almost all cells in the scDNA-seq are identical and diploid? I think some bootstrapping approach (leave one out etc) would be helpful to show that a definitive match cannot be made.

4. Another approach to bootstrapping could be used by exploiting the heterogeneity of BE data encoded in patient IDs. A quick glance at the CNV profiles in the original study shows that these profiles are patient-specific. The authors could benchmark the ability of theirs and other tools to recover scRNA-seq patients' labels after modality integration. I would assume that after integration, scRNA-seq data that originates from dysplastic samples with CNV should be accurately assigned to the correct patient, and cells without CNV changes would probably be randomly distributed.

REVIEWER COMMENTS

Reviewer #1 (Remarks to the Author): Expert in single-cell multi-omics, bioinformatics, cancer genomics and evolution

This manuscript introduces MaCroDNA, a computational method to map unpaired scRNA-seq and scDNA-seq data. The method performs a simple correlation-based association between cells from both technologies. The authors apply it to one data set containing both measurements from the same cells for validation and to another with unpaired data for exploratory analysis.

MaCroDNA finds the mapping between cells from scRNA-seq and cells from scDNA-seq that maximises the sum of the Pearson correlations between each cell's gene expression and its copy number profile. This is a simple approach that is akin to the baseline method used in the simulations for the clonealign [1] paper, and which is outperformed by clonealign in that work. However, in the present manuscript, clonealign performs poorly and MaCroDNA performs very well in the real data set with ground truth. It is unclear why clonealign performs so poorly, completely failing in some cases where MaCroDNA obtains a high accuracy. Additionally, it is unclear whether the results in this ground truth data set apply to unpaired data, which is obtained by different protocols and usually yields a much larger number of cells from both modalities. For users to decide which method to use, the authors must explore the differences between the methods that explain their performance discrepancies, and ideally apply them also in simulated data.

In summary, the authors introduce a simple method that requires more careful validation and comparison with the state-of-the-art. The manuscript is well-written. We detail our concerns below.

Major comments

Our main concern is the comparison with clonealign, which the authors indicate as the state-of-the-art in unpaired scRNA-seq/scDNA-seq data mapping. MaCroDNA consistently outperforms clonealign in 3 real data sets and with multiple levels of

resolution of the scDNA-seq data. All 3 data sets have at least 2 very distinct copy number clones (Fig. S5 of [2]), which makes it hard to understand why one data set in Fig S2 with agglomerative clustering yields a 0% accuracy in clonealign, and with intNMF all the data sets lead to that performance (Fig S2 c).

We elaborated on this comment and the third major comment together (see below).

The results become even more puzzling when the performance of clonealign improves if the input copy numbers are log-transformed, even though the clonealign model assumes actual copy numbers as input.

We thank the reviewer for this comment. By "log-transformed," we refer to the input data used for obtaining clusters (clones), not the input data for clonealign, Seurat, or MaCroDNA. Our aim was to mitigate the influence of clones on integration accuracy. Therefore, we employed two different clustering methods, namely agglomerative clustering, and intNMF. For each clustering method, we used either untransformed or log-transformed data to obtain the clusters.

In our experiments, we used actual copy numbers (absolute integer copy number values) as the input for clonealign. We preprocessed the copy number data based on clonealign's original study, and you can find the details in Section 3.2 of our paper titled "Preprocessing on CRC data set."

The authors must elaborate on the results presented in Figure S2, in which clonealign consistently gets 0% accuracy on the CRC04 patient regardless of the number of clones. This figure is puzzling because clonealign assigns 57 cells (i.e. all cells) to CRC04_clone1 that actually belong to CRC04_clone0 in the "Reference" (which we assume means the ground truth). But if all cells in the reference belong to only one cluster, why are there two clusters being given to clonealign? What is CRC04_clone1?

Figure S2 illustrates the result of clonealign at different resolutions. The clones (clusters) depicted in Figures S2a-f were merged from the clones in Figure S2g. For instance, crc04_clone0 in Figure S2c is the combination of crc04_clone0 and crc04_clone1 in Figure S2g, while crc04_clone1 in Figure S2c is the merged result of crc04_clone2 and crc04_clone3 from Figure S2g.

In Figure S2g, clonealign assigned all cells to crc04_clone2, resulting in all cells being allocated to crc04_clone1 in Figure S2c.

In Figure S2c, the ground truth reveals that crc04_clone0 comprises 57 cells, while crc04_clone1 has 0 cells. This discrepancy is also caused by Figure S2g. Figure S2g displays four clones, with crc04_clone0 containing 33 cells, crc04_clone1 containing 24 cells, and both crc04_clone2 and crc04_clone3 containing 0 cells, as shown in the figure. The reason for the 0 cell counts is that, for cluster generation and evaluation of integration methods, we included all cells, including those with both RNA and DNA, those with only RNA, and those with only DNA. However, during the evaluation of integration accuracy, only cells with both RNA and DNA are

utilized. This results in the 0 cell count for crc04_clone2 and crc04_clone3 in Figure S2g, subsequently causing the 0 cell count for crc04_clone1 in Figure S2c.

Taken together, these results warrant more validation. This can include exploring what clonealign assignments are incorrect or performing simulation studies that try to recover situations in which MaCroDNA and clonealign behave more comparably, as they in principle should. Indeed, as clonealign is state-of-the-art, reporting such extremely poor performance – contradicting the original publication – asks for a closer look at the results.

Regarding the performance of clonealign, it exhibits its worst performance when using clusters from the agglomerative clustering method and log-transformed data, as depicted in Figure S9. To investigate the reasons behind this, we conducted extensive experiments and discovered that clonealign strongly favors clones, where the most common copy number constitutes a significantly high proportion. You can find more detailed information in Supplementary Section 2, titled "Analysis of clonealign performance."

Minor comments

When explaining the agglomerative clustering step to define the number of clones, the authors confusingly describe a procedure that starts by inferring 4 clones per patient independently and then merging the total of 12 clusters across the 3 patients until 2 clusters per patient are obtained. It is confusing that they describe the merging step starting from all 12 clusters across patients, instead of doing it per patient. Please re-write for clarity.

To clarify our merging scheme, we rewrote the description of this procedure in subsection 3.5, named "Clonal inference by agglomerative clustering," under the Methods section of the main text.

Relatedly, the figures in the supplement with confusion matrices makes it seem as if the authors ran each method on all 3 patients simultaneously. This would probably hurt Seurat's and clonealign's performance due to the resulting batch effects.

We ran each method on each patient separately. To avoid this confusion, we added some sentences in the captions of all figures with confusion matrices in the supplement to clarify that the results of each patient are obtained independently from the others (in blue).

Please label the dots in each panel by the data set (patient) of origin.

To annotate the results based on the patient of origin, we used different shapes for the dots coming from different patients. Now, the colors differentiate the methods, and marker shapes differentiate the patients (see Figures 2 and 3 in the main text).

In their clonealign script on Github, they do ``dna_data[dna_data == 0] <- 1`` to avoid NaNs. But, if dna_data contains CNVs, they should set it to a very low number instead of 1. Setting CNVs of 0 to 1 may have a strong effect on the resulting assignments.

We have corrected the preprocessing of CNVs for clonealign using two approaches. One approach was the removal of the genes with at least one zero copy number value from the data, according to one of the issues posted on the clonealign Github repository at

<https://github.com/kieranrcampbell/clonealign/issues/4>.

The second approach to preprocessing the CNV inputs was to add the pseudo-count value of 1 to all CNVs before feeding them into clonealign. We applied both preprocessing approaches. Since the removal of genes with zero copy numbers yielded better results, we replaced clonealign's results in the main text with the results obtained from this preprocessing procedure (Figures 2 and 3 in the main text). To demonstrate the difference between the two preprocessing schemes on the accuracy of clonealign, we reported the results of both preprocessing schemes in the SI under the subsection named "Investigating clonealign's accuracy by adding pseudo-count values to the inputs". In this subsection, we included a figure showing the results of clonealign tagged with "clonealign_rm0" and "clonealign_add1" for the first and second approaches, respectively.

From the data on Github, we can see that some genes have huge copy numbers across all cells (e.g. 30). If the average CNV profile across cells in the same cluster is taken as clonal CNVs, and these cells are in the same cluster as cells with a much lower ploidy, it will be difficult to perform a correct assignment because the average will not represent any cell very well.

To have a better representation of the CNV profiles in a clone, we replaced the average of CNVs by the median to obtain the clonal copy number profile as suggested in the clonealign paper. The results of clonealign are now updated with the new preprocessing approaches (described above) and the use of the median for obtaining the clonal CNV profiles.

The authors mention and cite another method to perform scRNA/scDNA integration, CCNMF. They might also want to mention SCATrEx [3], which has the same purpose (with additional features).

We thank the reviewer for bringing the SCATrEx paper to our attention. We have now cited this paper and added some text describing this method in the Introduction in the main text (in blue).

In Section 3.8, why is the gene expression counts matrix G defined in the set of integer numbers? Don't the authors use either raw counts (positive integers) or log-normalized counts (reals)?

We have corrected the description of the gene expression counts matrix G in section 3.8 by adding the point that the matrix could be either non-negative integers or real values in case of log-transforming the gene expressions and removing the notations for integer or non-negative integer values from the matrix notations.

In Section 3.8.1, what is the difference between A and I ?

The A variable is the binary correspondence matrix whose entries are the I_{ij} 's that are the indicator variables for correspondence of each pair of scRNA-seq and scDNA-seq cells. For a more convenient notation, we changed the A variable to bold I , (I) so that it better represents the matrix of all I_{ij} 's.

Throughout the text, it is unclear whether the input data from the scDNA modality is read counts or copy number values obtained from some other method. In Section 1, they first say that the inputs to MaCroDNA include "scDNA-seq copy number values", but then say that "these measurements are provided (...) in the form of counts matrices of single cells from the same tissue". In Section 3.8.3, they state that they use the "raw read counts in the CNA profile". Please clarify this aspect.

We clarified the definition of inputs by adding or changing the sentences in section 1.1 (Overview of MaCroDNA), in the caption of Fig.1, and in section 3.8.3. The inputs of MaCroDNA are the gene expression read count tables (or their log-transformed values) and the scDNA-seq absolute copy numbers (or their log-transformed values). We made sure that this is now consistent throughout the main text.

"We then used MaCroDNA to assemble a set of virtual cells; each virtual cell merged one cell from the scDNA-seq data set, and one cell from the scRNA-seq data set, based on the one-to-one correspondences identified by MaCroDNA." – Please clarify what a virtual cell is and how MaCroDNA obtains them.

By a *virtual cell*, we meant a pair of RNA-DNA cells identified by MaCroDNA that is treated as a single cell for which we have both RNA and DNA information. To avoid this confusion and convey the message better, we replaced the term *virtual cell* with a descriptive sentence: "We, then, treated each matched pair of RNA-DNA cells identified by MaCroDNA as a single cell for which we have both RNA and DNA information" in blue text in the same section.

The colors in Figure 5 are difficult to distinguish.

We changed the coloring of Figure 5, and the thickness of the curves so that they are more visible and distinguishable.

What is the reason behind the unusual scale in the numbers of cells in Figure 5?

The unusual scale in the numbers of cells in Figure 5 is due to the way we created different-sized datasets to evaluate the computational cost. We randomly selected some cells from each patient's original data and repeated this process 10 times for each method and size. The sizes ranged from half to the full original size, with increments of 10% of the full original size. Since the original data has 835 cells, this resulted in datasets with 418, 501, 584, 668, 752, and 835 cells.

References

[1] Campbell, Kieran R., et al. "clonealign: statistical integration of independent single-cell RNA and DNA sequencing data from human cancers." *Genome biology* 20.1 (2019): 1-12.

[2] Bian, Shuhui, et al. "Single-cell multiomics sequencing and analyses of human colorectal cancer." *Science* 362.6418 (2018): 1060-1063.

[3] Ferreira, Pedro F., et al. "Mapping single-cell transcriptomes to copy number evolutionary trees." *Research in Computational Molecular Biology: 26th Annual International Conference, RECOMB 2022, San Diego, CA, USA, May 22–25, 2022, Proceedings*. Cham: Springer International Publishing, 2022.

Reviewer #2 (Remarks to the Author): Expert in gastroesophageal cancer genomics, Barrett's oesophagus genomics, and single-cell genomics

The manuscript presented by Edrisi and colleagues aims to implement a simple method for the integration of single-cell RNA and DNA (defined as copy number alteration profiles) data. To achieve this goal, the authors developed MaCroDNA. This technique uses Pearson correlation to identify a relationship between expression counts (scRNA-seq) and CNV data obtained using scDNA-seq method. The method works in the space of gene-by-cell matrix. It also aims to identify at most one cell with scRNA-seq data that correspond to one scDNA-seq cell. The authors used two publicly available datasets from single cell studies of colorectal cancer and Barrett's esophagus. Only colorectal cancer study had matched DNA and RNA data. They benchmarked their method against other tools, and they show that they tend to outperform these tools in accuracy.

The manuscript is well presented with a clear message and well-described statistical methods. The available code is well described. As the author mentioned

in the discussion, their approach is “simplistic”, as a result, the tool might be missing important functionality. The following question stem from the fact that cells might be misassigned in specific conditions and I would suggest that assignment of confidence scores to the matches should be a priority.

1. Firstly, the key assumption of this tool is that the scRNA-seq and scDNA-seq datasets originate from the same populations and the distribution of cell types should be similar across the modalities. However, single-cell approaches are intrinsically noisy methods where sample preparation is a key component for good recovery of cell type diversity (e.g. PMID: 32487174). As a result, the assumption that all cell types and their proportion across modalities will most likely not hold true in most settings. Could the authors elaborate on cell type assignment when there is an in-balance in cell populations across modalities? For example, they could bootstrap colorectal data by removing scDNA-seq from the cells with match RNA and DNA data to test if the RNA component is still assigned to the correct cluster.

As suggested, we measured the effect of the removal of the scDNA-seq cells with joint RNA-DNA information on the clonal assignment accuracy of their scRNA-seq pairs in the CRC data. In particular, for each CRC patient, we randomly selected 10 scDNA-seq cells with both RNA and DNA information, removed them from the DNA data, and ran MaCroDNA. We repeated this experiment 10,000 times with different random seeds and measured the clonal assignment accuracy of the true scRNA-seq pairs. We performed this experiment on all patients with different clustering techniques for clonal inference. The results show that the accuracy decreases in the majority of cases, with a median ranging from 0% to 10%. The figures and detailed description of the experiment are written in blue text in the SI under section 3.1.1, titled “Random removal of scDNA-seq cells from the DNA data.”

2. Similarly, the method aims to assign only one cell from the scRNA-seq data set to exactly one cell from the scDNA-seq cell. Also, only one scRNA-seq can be assigned to each scDNA-seq cell. Could the author elaborate on how would the method behave in the following conditions that can arise due to sample preparation discrepancies: RNA: 10 cells in population A and 90 cells in population B, DNA: 90 cells in population A and 10 cells in population B. Surely, since the method aims to perform 1-to-1 matching, many cells will be assigned to wrong populations?

To study the effect of deviation from the original clonal proportions on the performance of MaCroDNA, we randomly drew new clonal proportions in the DNA data from a Dirichlet-multinomial distribution and sampled the scDNA-seq cells with replacement within each clone to achieve the drawn proportions. Then, we inputted the randomly sampled scDNA-seq cells along with the original RNA data to MaCroDNA and measured the accuracy of clonal assignments for the scRNA-seq

cells for which we know the true scDNA-seq pairs. We repeated this test 10,000 times and reported the distribution of the accuracy values, as well as the scatter plots showing the relationship between the accuracy and the deviation from the original clonal proportions (measured as the Earth mover's distance). Our results showed to what extent the accuracy decreases and how much it depends on the choice of clustering algorithm for clonal inference. The data also showed a strong negative Spearman correlation between the deviation from the original proportions and the accuracy. The figures and description of this experiment are reported in blue text in SI under section 3.1.2, named "Resampling clonal proportions in DNA data."

3. In the BE dataset, the authors used NDBE and normal samples that have few copy number changes. However, as the original study shows, there are many cell populations that can be identified using scRNA-seq. Could the authors elaborate on the stability of data integration when almost all cells in the scDNA-seq are identical and diploid? I think some bootstrapping approach (leave one out etc) would be helpful to show that a definitive match cannot be made.

To assess the confidence in the assignments, we performed two following experiments:

a) Comparing the results of random assignments of RNA cells to DNA cells in BE biopsies. For each biopsy, we performed the random assignments 100 million times and calculated the sum of the Pearson correlation coefficients of the paired cells. We plotted the histogram of the scores of the random assignment along with the score from MaCroDNA. This random assignment test showed that none of the random assignments could reach the score of MaCroDNA, showing that MaCroDNA's results are non-trivial. These results are available in SI under section 3.2, named "Random assignment test for BE biopsies," written in blue text.

b) The second experiment was performed to assess the stability of the integration, as the reviewer suggested. For each biopsy, and in each round of this experiment, we leave out one cell at a time (here, we leave out one RNA cell) from the inputs, we run MaCroDNA, and study the effect of this perturbation on the assignments compared to the assignments obtained by running MaCroDNA on the entire set of RNA cells. Specifically, for each RNA cell g , we collected the unique DNA cell IDs that g was assigned to during these leave-one-out trials. Among these unique cell IDs, we counted the number of cell IDs that were different from g 's assigned DNA cell when running MaCroDNA on the entire set of RNA cells. This gave us a measure of the diversity of DNA assignments for all RNA cells that we named "assignment instability index" (AII). To investigate the effect of heterogeneity on the AIs, we used the median of the L1-norm pairwise distances between DNA cells as our heterogeneity score for each BE biopsy. Next, we drew the regression plot between the biopsies' heterogeneity scores and their maximum AIs. The negative slope of the plot and the strong negative correlation values show that as the

heterogeneity increases, the assignment instability index decreases, and MaCroDNA can reach more definitive matches.

The details of this experiment are written in the SI, under section 3.3, titled “Stability analysis of MaCroDNA’s assignments for BE biopsies.”

4. Another approach to bootstrapping could be used by exploiting the heterogeneity of BE data encoded in patient IDs. A quick glance at the CNV profiles in the original study shows that these profiles are patient-specific. The authors could benchmark the ability of theirs and other tools to recover scRNA-seq patients’ labels after modality integration. I would assume that after integration, scRNA-seq data that originates from dysplastic samples with CNV should be accurately assigned to the correct patient, and cells without CNV changes would probably be randomly distributed.

The reviewer raises a great question. We designed an experiment to assess the ability of our method to recover the scRNA-seq patients’ labels after modality integration and investigate the effect of heterogeneity level on the results. The experiment is as follows:

We aggregated all scDNA-seq cells from all DNA biopsies into one unifying data set, **C**. Similarly, we pooled all scRNA-seq cells from all biopsies into one set, **G**, and ran MaCroDNA to assign cells from **C** to **G**. Given the assignments, each scRNA-seq cell, **g**, from **G**, is assigned accurately to a scDNA-seq cell, **c**, from **C** if the biopsy label of **g** is the same as the biopsy label of **c**.

We measured the accuracy of assignments per biopsy, as the number of the scRNA-seq cells that are assigned to a scDNA-seq cell from the same biopsy divided by the total number of scRNA-seq cells in the biopsy.

Among the BE biopsies, all healthy and non-dysplastic biopsies displayed very low accuracy while one of the low-grade, two of the high-grade, and the cancer biopsies demonstrated high accuracy values. This observation suggests that the definitiveness and confidence of the assignments increase with the level of heterogeneity in the biopsies. Moreover, we looked into the distribution of each biopsy’s scRNA-seq cells’ assignments to the same biopsy and the other. We observed that the distributions of the healthy and non-dysplastic Biopsies are more uniform, while in most low-grade, high-grade, and cancer biopsies, the assignment counts to the same biopsy are much higher, and distribution is more concentrated around the same biopsy. The figures and description of this experiment are provided in the SI, under section 3.4, named “Retrieval of BE biopsy labels” in blue text.

REVIEWER COMMENTS

Reviewer #1 (Remarks to the Author):

The authors have mostly addressed our concerns. The method they propose seems to perform better than the state-of-the-art methods for cell-to-clone assignment from unmatched single-cell DNA and single-cell RNA data, in the data sets they considered. We still believe some more validation is required. We detail our comments below.

Our main concern regarding the low performance of clonealign in Figure S2 has been partly addressed. The authors have added details to the Methods section describing the agglomerative clustering results with different resolutions. They define clones with agglomerative clustering and apply each method at different resolutions. They provide the methods with all the clones from all the modalities, and then report the results using only the matched data.

The authors added Supplementary Section 2 which extensively addresses clonealign's generally poor performance on their data. This is a welcome addition to the manuscript. Interestingly, clonealign tends to assign all or most cells to the clone with the most uniform copy number, and this is especially bad when a neutral clone is present. In principle, the presence of a neutral clone would help identify the other clones more accurately, but the authors observe the opposite. The authors find that the clonealign results are mostly determined in the first iteration of its optimization procedure, which suggests an issue with the optimization.

In order to understand whether the clonealign behaviour is due to a computational issue or something more fundamental in the way the method works, it would be important to report whether the authors could replicate the original results in the clonealign publication.

Additionally, demonstrating the performance of a simulation study covering a wider range of copy number profiles would make the authors' points stronger. Is this performance gain only visible on the new data the authors show, or can users expect MaCroDNA to be better in general?

The authors have also addressed our minor requests satisfactorily and clarified the text according to our comments.

The panels on the supplementary figures should be labelled.

The authors mention in their rebuttal that the clones with 0 cells in Figure S2 in the ground truth are due to using both the matched and the unmatched cells to define clones in the first place, and then using only the matched cells for evaluation. This should be noted in the text or in the caption.

Reviewer #2 (Remarks to the Author):

The authors have addressed my prior comments and concerns by performing additional data analysis. In particular, they performed requested simulations of in-balance between scRNA-seq and scDNA-seq populations with particular attention on the CRC samples and in the additional analysis of BE samples they identified that their tool cannot clearly assign scRNA information to scDNA information if the population of cells is uniform on DNA level. However, despite this significant additional analysis, the authors did not address these results in the main text of the manuscript. The new supplementary text is not referenced in the main body of the article, nor are the results obtained in response to concerns raised by both reviewers. The new evidence should be included in the results and discussion, with a particular focus on decreased accuracy of the tool in highly homogeneous samples (as demonstrated during the reanalysis of CRC and BE samples). This effect does not seem to be a limitation of the tool as the biological differences drive these issues. The issues arising with clonealign analysis identified in response to reviewer 1 comments should also be included in the main text (potentially discussion).

Minor comments:

1. Could the author comment on the source of a very low score in the random assignment calculations presented in Figure S28 for normal and non-dysplastic BE samples? The subsequent analysis seems to indicate that the results are unstable for these samples due to high homogeneity. Since the score is calculated as a sum of Pearson correlations, the true score might be pushed towards higher values by a small number of outlier cells (potentially identified in Figure S29). Would the result in Figure S28 still hold if the authors chose the median Pearson correlation for each comparison rather than the sum?
2. The analysis presented in section 3.4 demonstrated well the importance of heterogeneity in the scDNA-seq data for accurate cell assignment. I would strongly suggest that this information should be presented in the main body of the manuscript as it demonstrates the importance of this feature in future applications of the method. It further highlights the strength of the authors' methodology by focusing on real, diverse samples. In the authors have information about the identity of scRNA-seq cells (tumor vs normal epithelium vs immune/stromal cells), the authors can further improve this analysis by stratifying the cells into tumor/non-tumor groups. It should be assumed that tumor cells should be ones that are accurately assigned to the current patient.
3. Line 116: BE is defined as "an esophagus in which any portion of the normal distal squamous epithelial lining has been replaced by metaplastic columnar epithelium, which is clearly visible endoscopically (≥ 1 cm) above the GEJ and confirmed histopathologically" (PMID: 21376940). This epithelium is not fully intestinal in nature. The authors can correct this statement to: "BE is a metaplasia, presumed to be caused by gastroesophageal reflux disease, where squamous epithelium of the esophagus is replaced by metaplastic epithelium with intestine-like goblet cells"
4. Lines 123-125: It is known that large genetic changes (CNV) are good predictors of progression (PMID: 27538785, 32895572, 34545238)

Reviewer #1 (Remarks to the Author):

The authors have mostly addressed our concerns. The method they propose seems to perform better than the state-of-the-art methods for cell-to-clone assignment from unmatched single-cell DNA and single-cell RNA data, in the data sets they considered. We still believe some more validation is required. We detail our comments below.

Our main concern regarding the low performance of clonealign in Figure S2 has been partly addressed. The authors have added details to the Methods section describing the agglomerative clustering results with different resolutions. They define clones with agglomerative clustering and apply each method at different resolutions. They provide the methods with all the clones from all the modalities, and then report the results using only the matched data.

The authors added Supplementary Section 2 which extensively addresses clonealign's generally poor performance on their data. This is a welcome addition to the manuscript. Interestingly, clonealign tends to assign all or most cells to the clone with the most uniform copy number, and this is especially bad when a neutral clone is present. In principle, the presence of a neutral clone would help identify the other clones more accurately, but the authors observe the opposite. The authors find that the clonealign results are mostly determined in the first iteration of its optimization procedure, which suggests an issue with the optimization.

In order to understand whether the clonealign behaviour is due to a computational issue or something more fundamental in the way the method works, it would be important to report whether the authors could replicate the original results in the clonealign publication.

To replicate the results of the original study, we attempted to run clonealign at the beginning of this study (spent a few months) and for the second time, during the three months that we worked on the previous revision. Despite all our efforts, running clonealign on the original data was extremely challenging due to the numerous errors we encountered especially when obtaining the clone-specific copy number profiles, and unfortunately, we were unable to run it to reproduce the original results. However, we tried our best to investigate its behavior as thoroughly as possible on the CRC data set we readily had. In the process of diagnosing clonealign's behavior on the CRC data set—as mentioned in the previous response letter—we determined the issue to be caused by the optimization procedure (please refer to Section Analysis of clonealign performance in Supplementary Materials).

Later, to understand whether the root cause is a numerical (computational) issue or comes from the original variational inference modeling (more fundamental) we realized that we would need to fully understand the ELBO function of clonealign. This effort, however, was hindered by the lack of a detailed description of the ELBO in the original paper and even reading the clonealign's codes line by line did not help us. Unfortunately, we could not proceed further from this point.

Additionally, demonstrating the performance of a simulation study covering a wider range of copy number profiles would make the authors' points stronger. Is this performance gain only visible on the new data the authors show, or can users expect MaCroDNA to be better in general?

Although we agree that a simulation study can be very insightful, we would like to emphasize that this is extremely challenging because generating paired DNA/RNA data is not a trivial task; it's akin to figuring out much of the central dogma of molecular biology. In fact, a major novel aspect of our manuscript is that it's the first study to assess the performance of methods on an empirical data set where the ground truth (DNA/RNA match) is known. One could of course write a simulator to generate such data, but it would be very far from biological realism and very poor compared to ground truth biological data like we have used here.

The authors have also addressed our minor requests satisfactorily and clarified the text according to our comments.

The panels on the supplementary figures should be labelled.

We have labeled all the panels on the supplementary figures.

The authors mention in their rebuttal that the clones with 0 cells in Figure S2 in the ground truth are due to using both the matched and the unmatched cells to define clones in the first place, and then using only the matched cells for evaluation. This should be noted in the text or in the caption.

We have added the following sentence to the main text for clarification (Lines 243-246):

"It should be noted that in all experiments related to cell-to-clone assignment and prediction of clonal prevalences (including the results in Supplementary Materials, Figures S1--S19), we applied clustering algorithms to all DNA cells, but only used the cells with both DNA and RNA data for accuracy measurement."

Reviewer #2 (Remarks to the Author):

The authors have addressed my prior comments and concerns by performing additional data analysis. In particular, they performed requested simulations of in-balance between scRNA-seq and scDNA-seq populations with particular attention on the CRC samples and in the additional analysis of BE samples they identified that their tool cannot clearly assign scRNA information to scDNA information if the population of cells is uniform on DNA level.

However, despite this significant additional analysis, the authors did not address these results in the main text of the manuscript. The new supplementary text is not referenced in the main body of the article, nor are the results obtained in response to concerns raised by both reviewers. The new evidence should be included in the results and discussion, with a particular focus on decreased accuracy of the tool in highly homogeneous samples (as demonstrated during the reanalysis of CRC and BE samples). This effect does not seem to be a limitation of the tool as the biological differences drive these issues. The issues arising with clonealign analysis identified in response to reviewer 1 comments should also be included in the main text (potentially discussion).

Thanks for your suggestion. Here, we detail the changes we have made in the main text to include the results and findings from the supplementary materials.

- The results from Section Resampling and stability analyses of Supplementary Materials were referenced in the Results (Lines 282-287, 311-318) and Discussion (Lines 447-453) Sections of the main text. As suggested, we focused on decreased stability and definiteness of the assignments in highly homogeneous biopsies as this information is important for the future application of the tool by the users.
- The second preprocessing method (adding a pseudo-count value of 1 to all absolute copy numbers) that we tried on clonealign was not referenced in the main text. For this, we added sentences in the Methods (Lines 486-489) and Results (Lines 258-262) Sections referencing the corresponding section and figure in Supplementary Materials.
- Since figures S1–S18 were only referenced in the Methods Section of the main text, we referenced them in the Results Section explaining they illustrate the results of different methods under various scenarios for data preprocessing, clustering techniques, and clustering resolutions (Lines 262-266).
- Regarding the issues arising with clonealign, we have included a new paragraph in the Discussion Section of the main text (Lines 426-432).

Minor comments:

1. Could the author comment on the source of a very low score in the random assignment calculations presented in Figure S28 for normal and non-dysplastic BE samples? The subsequent analysis seems to indicate that the results are unstable for these samples due to high homogeneity. Since the score is calculated as a sum of Pearson correlations, the true score might be pushed towards higher values by a small number of outlier cells (potentially identified in Figure S29). Would the result in Figure S28 still hold if the authors chose the median Pearson correlation for each comparison rather than the sum?

Thanks for bringing up this important point about the effect of outlier scores. We first, repeated the random assignment experiment (with the same random seed to reproduce the same results as in the previous experiment), and used the median of Pearson correlation coefficients between the paired cells as the score of the assignments (for both MaCroDNA and random assignments). Interestingly, the new results demonstrated the same pattern, i.e., the ranking of patients in terms of random assignment medians is the same as in the previous figure we generated for the sum of Pearson correlations (see Figure S29 in the supplementary materials). We observed that MaCroDNA's medians are still far from those of the random assignments with approximately zero p-values. Moreover, the normal and non-dysplastic BE samples still have lower scores compared to the high-degree and cancer biopsies. To better understand the difference between the sum and median of Pearson correlations from random assignments, we drew a regression plot as follows:

For each BE biopsy, and for each score (sum or median), we calculated the median of random assignment scores as the representative of the random assignment scores' distribution for that biopsy and score. This yielded two vectors, **M** and **S** for median and sum, respectively. Next, we calculated the Pearson and Spearman correlation coefficients between **M** and **S**. Figure S30 in the supplementary materials shows a strong and significant correlation between the results of the two scores, indicating that the original distribution of Pearson correlation values in the random assignments was not skewed by the outliers.

As you mentioned and Figure S30 clearly shows, the normal and non-dysplastic biopsies demonstrate lower scores in their random assignments compared to the other biopsies. Although the level of heterogeneity in these biopsies seems to be a contributing factor, here, we think that heterogeneity does not affect a random assignment scheme as this scheme does not prioritize the assignment of cells with higher correlation. To have a higher correlation in random assignments, there must be copy number changes shared by a large portion of DNA cells that drive differential gene expressions (such CNAs might have occurred at the early stages of clonal expansion and disease development). Under this scenario, random assignments show high correlations even if the biopsy is homogeneous. Therefore, we speculate that copy number changes present in major DNA clones might be the

key factor (the analysis of the clone-specific CNAs in high-degree and cancer biopsies in Figures 2 and 3 from the original study can be pursued in this regard). Of course, our hypothesis requires more investigation and we leave it as future direction.

The above statements are provided in the supplementary materials (Lines 230-268) and are referenced in the main text (Lines 311-314).

2. The analysis presented in section 3.4 demonstrated well the importance of heterogeneity in the scDNA-seq data for accurate cell assignment. I would strongly suggest that this information should be presented in the main body of the manuscript as it demonstrates the importance of this feature in future applications of the method. It further highlights the strength of the authors' methodology by focusing on real, diverse samples. In the authors have information about the identity of scRNA-seq cells (tumor vs normal epithelium vs immune/stromal cells), the authors can further improve this analysis by stratifying the cells into tumor/non-tumor groups. It should be assumed that tumor cells should be ones that are accurately assigned to the current patient.

We presented the findings from Section 3.4 of the Supplementary Materials in the main body of the manuscript in the Results (Lines 314-318) and Discussion (Lines 449-453) Sections.

Unfortunately, we don't have the information about the identity of scRNA-seq cells to improve the analysis according to the reviewer's suggestion.

3. Line 116: BE is defined as "an esophagus in which any portion of the normal distal squamous epithelial lining has been replaced by metaplastic columnar epithelium, which is clearly visible endoscopically (≥ 1 cm) above the GEJ and confirmed histopathologically" (PMID: 21376940). This epithelium is not fully intestinal in nature. The authors can correct this statement to: "BE is a metaplasia, presumed to be caused by gastroesophageal reflux disease, where squamous epithelium of the esophagus is replaced by metaplastic epithelium with intestine-like goblet cells"

We have corrected this statement accordingly (Lines 115-117 in the main text).

4. Lines 123-125: It is known that large genetic changes (CNV) are good predictors of progression (PMID: 27538785, 32895572, 34545238)

Thanks for your comment. We have added the sentence "*This result aligns with the previous studies suggesting that the copy number changes are good predictors of progression from Barret's esophagus to esophageal adenocarcinoma*" along with citing the above references (Lines 125-127 in the main text).

Also, we changed the abstract so that it conveys the message that our finding is in agreement with the previous studies (Lines 14-17 in the main text).

REVIEWERS' COMMENTS

Reviewer #1 (Remarks to the Author):

The authors have further clarified the reasons for the low performance obtained with clonealign. However, their response to our suggestion for adding a simulation study is unsatisfactory, as we explain below.

We agree that simulated paired single-cell DNA/RNA data most likely does not capture all the complexity of real data – but that is not the point of a simulation study. The authors present a method that makes assumptions about how these data types are related, and only a simulation study can accurately show how their method performs under data that agree with these assumptions, data that does not, and anything in between. It also provides a better picture of how it compares with other methods. Despite the difficulties that the authors had with running the clonealign method, the simulation study presented in that publication is a good example of how to evaluate the performance of a model as its main assumptions are tested.

We also agree with the authors' point that their study has a significant benefit over the current literature due to their evaluations on real data with ground truth. However, we believe that presenting a computational method and showing results on two real data sets does not inform users about when to expect the method to work or fail, and the reasons why it would. Presenting some sort of simulation study is standard practice in bioinformatics.

Reviewer #2 (Remarks to the Author):

The authors have addressed all outstanding issues.

Reviewer #3 (Remarks to the Author): Arbitrating Reviewer with expertise in single-cell DNA and RNA analysis and integration, bioinformatics, statistics, cancer evolution and heterogeneity

EDITORIAL NOTE: This Reviewer only provided confidential remarks to the Editor. This Reviewer considers that there could have been a misunderstanding regarding the requests about simulations, as these could have been about the modulation of clonal prevalence. However, they do consider that the use of real scTrio-seq data was the best option for benchmarking in this study, and that the suggested simulations would not add much. Therefore, the initial concerns were addressed.

REVIEWERS' COMMENTS

Reviewer #1 (Remarks to the Author):

The authors have further clarified the reasons for the low performance obtained with clonealign. However, their response to our suggestion for adding a simulation study is unsatisfactory, as we explain below.

We agree that simulated paired single-cell DNA/RNA data most likely does not capture all the complexity of real data – but that is not the point of a simulation study. The authors present a method that makes assumptions about how these data types are related, and only a simulation study can accurately show how their method performs under data that agree with these assumptions, data that does not, and anything in between. It also provides a better picture of how it compares with other methods. Despite the difficulties that the authors had with running the clonealign method, the simulation study presented in that publication is a good example of how to evaluate the performance of a model as its main assumptions are tested.

We also agree with the authors' point that their study has a significant benefit over the current literature due to their evaluations on real data with ground truth. However, we believe that presenting a computational method and showing results on two real data sets does not inform users about when to expect the method to work or fail, and the reasons why it would. Presenting some sort of simulation study is standard practice in bioinformatics.

We have added text to the Results and Discussion sections of the main text pointing out that a simulation study can very insightful (in agreement with the reviewer's opinion) but our knowledge is too limited to design a realistic simulator. Specifically, we added the following to lines 165-166 in the Results section (in blue text):

“Given the limited ability to produce realistic synthetic data, this study assesses the accuracy of methods for integrating DNA and RNA data using this empirical data set.”

Also, in the Discussion section we added the following text to lines 404-407 (in blue text):

“While simulation studies based on synthetic single-cell gene expression and copy number data can be very insightful for the evaluation of integration tools, such a

task would be very challenging as our knowledge of the DNA-to-RNA process is too limited to produce realistic simulations.”

We thank the reviewer for great suggestions that helped us improve the quality and clarity of our manuscript.

Reviewer #2 (Remarks to the Author):

The authors have addressed all outstanding issues.

We would like to express our gratitude to the reviewer for the valuable suggestions that helped us improve the quality of our study.

Reviewer #3 (Remarks to the Author): Arbitrating Reviewer with expertise in single-cell DNA and RNA analysis and integration, bioinformatics, statistics, cancer evolution and heterogeneity

EDITORIAL NOTE: This Reviewer only provided confidential remarks to the Editor. This Reviewer considers that there could have been a misunderstanding regarding the requests about simulations, as these could have been about the modulation of clonal prevalence. However, they do consider that the use of real scTrio-seq data was the best option for benchmarking in this study, and that the suggested simulations would not add much. Therefore, the initial concerns were addressed.

We are glad that the arbitrating reviewer agrees with our concerns about the challenges of simulation studies in this area and that using the scTrio-seq data was the best option for benchmarking in our study.